# DFA-GNN: Forward Learning of Graph Neural Networks by Direct Feedback Alignment

**Gongpei Zhao**[1,2], **Tao Wang**[1,2]*, **Congyan Lang**[1,2], **Yi Jin**[1,2], **Yidong Li**[1,2], **Haibin Ling**[3]

[1]Key Laboratory of Big Data & Artificial Intelligence in Transportation, Ministry of Education, China
[2]School of Computer Science & Technology, Beijing Jiaotong University, China
[3]Department of Computer Science, Stony Brook University, USA
{csgpzhao, twang, cylang, yjin, ydli}@bjtu.edu.cn
hling@cs.stonybrook.edu

## Abstract

Graph neural networks (GNNs) are recognized for their strong performance across various applications, with the backpropagation (BP) algorithm playing a central role in the development of most GNN models. However, despite its effectiveness, BP has limitations that challenge its biological plausibility and affect the efficiency, scalability and parallelism of training neural networks for graph-based tasks. While several non-backpropagation (non-BP) training algorithms, such as the direct feedback alignment (DFA), have been successfully applied to fully-connected and convolutional network components for handling Euclidean data, directly adapting these non-BP frameworks to manage non-Euclidean graph data in GNN models presents significant challenges. These challenges primarily arise from the violation of the independent and identically distributed (*i.i.d.*) assumption in graph data and the difficulty in accessing prediction errors for all samples (nodes) within the graph. To overcome these obstacles, in this paper we propose **DFA-GNN**, a novel forward learning framework tailored for GNNs with a case study of semi-supervised learning. The proposed method breaks the limitations of BP by using a dedicated forward training mechanism. Specifically, DFA-GNN extends the principles of DFA to adapt to graph data and unique architecture of GNNs, which incorporates the information of graph topology into the feedback links to accommodate the non-Euclidean characteristics of graph data. Additionally, for semi-supervised graph learning tasks, we developed a pseudo error generator that spreads residual errors from training data to create a pseudo error for each unlabeled node. These pseudo errors are then utilized to train GNNs using DFA. Extensive experiments on 10 public benchmarks reveal that our learning framework outperforms not only previous non-BP methods but also the standard BP methods, and it exhibits excellent robustness against various types of noise and attacks.

## 1  Introduction

As a class of neural networks (NNs) specifically designed to process and learn from graph data, graph neural networks (GNNs) [Zhou et al., 2020, Wu et al., 2020] have gained significant popularity in addressing graph analytical challenges. They have demonstrated outstanding success in various applications, including recommendation systems [Wu et al., 2022], drug discovery [Xiong et al., 2021] and question answering [Yasunaga et al., 2021]. The impressive accomplishments of GNNs, as well as other neural network models, are largely attributed to the backpropagation (BP) algorithm [Hecht-Nielsen, 1992], which has emerged as the standard technique for training deep neural networks.

---

*Corresponding Author

38th Conference on Neural Information Processing Systems (NeurIPS 2024).

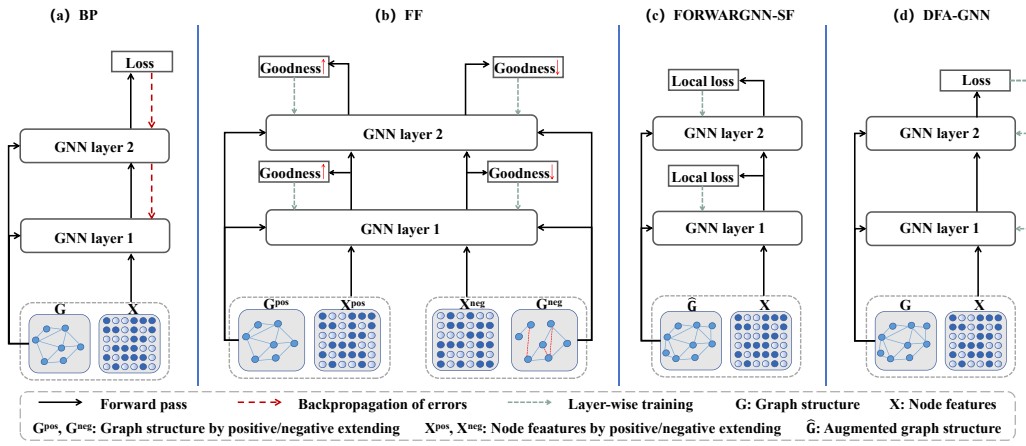

Figure 1: Illustrations of BP, FF, FORWARDGNN and proposed DFA-GNN.

The backpropagation algorithm adjusts neural network weights based on the loss between the prediction and the ground truth, and allows the network to learn and improve over time. However, despite its effectiveness, BP draws concerns on its biological plausibility for two main reasons [Hinton, 2022, Lillicrap et al., 2016]: (1) it uses the same weights in reverse order for both feedforward and feedback paths, creating the weight symmetry problem [Lillicrap et al., 2016]; and (2) its parameter updating relies on the activity of all downstream layers, leading to the update locking problem [Dellaferrera and Kreiman, 2022]. These limitations may as well impact the efficiency, scalability and parallel processing capabilities of neural network training.

To address these limitations, direct feedback alignment (DFA) [Nøkland, 2016] offers an effective alternative to BP by training neural networks through a single forward pass. DFA uses fixed random feedback connections to project output errors directly onto hidden neurons, allowing for parallel gradient computation and eliminating the need for sequential backward error propagation. While demonstrated a limited accuracy penalty compared with BP, DFA aligns with brain-like learning mechanisms through its use of global error modulation and local synaptic activity, making it a notable non-BP method applicable in areas such as image classification [Zhao et al., 2023] and privacy protection [Ohana et al., 2021].

Directly applying DFA to GNNs, however, faces two challenges: **(1)** graph data often violates the independent and identically distributed (*i.i.d.*) assumption and thus ties the supervision gradients with the graph structure, making the straightforward error projection of DFA inadequate; and **(2)** DFA requires the prediction errors of all the input samples, while for graph data, especially under the semi-supervised setting, samples (nodes) without ground truth meet problems for the error calculation, complicating the deployment of DFA to GNNs.

To tackle these challenges, in this paper we propose **DFA-GNN**, a non-BP learning framework tailored for graph neural networks. Our primary contribution is to improve and extend DFA to graph neural networks. Specifically, we redesign the random feedback strategy for graph data to make the DFA portable to GNNs. The information from the graph topology, in the form of an adjacency matrix, is incorporated into the feedback links to accommodate the non-Euclidean characteristics of graph data. We take graph convolutional network (GCN) [Kipf and Welling, 2016] as a case study, and derive the specific formula for updating parameters in each GCN layer. Furthermore, for the semi-supervised graph learning task, we develop a novel pseudo error generator that spreads residual errors from training data to generate a pseudo error for each unlabeled node. Such pseudo errors are then used for the training of graph neural networks by DFA.

In summary, our proposed learning procedure for GNNs contributes in three significant folds:

- We introduce DFA-GNN, a non-BP training algorithm that extends DFA to GNN architectures. It offers a more biologically plausible alternative to traditional BP methods.
- For semi-supervised graph learning tasks, we develop a novel pseudo error generator that propagates residual errors from the training data to create pseudo errors for unlabeled nodes.

- We prove the convergence of our DFA-GNN, and validate its effectiveness on 10 benchmarks. The experimental results demonstrate the superiority of our DFA-GNN against both traditional BP and the state-of-the-art non-BP approaches.

## 2 Related Work

The biological implausibility of BP mainly lies in weight transport and update locking issues. feedback alignment (FA) [Lillicrap et al., 2016] addresses the weight transport by using fixed random weights to convey error gradients. Building on this, direct feedback alignment (DFA) [Nøkland, 2016], direct random target projection (DRTP) [Frenkel et al., 2021] and PEPITA [Dellaferrera and Kreiman, 2022] further tackle the update locking problem with non-BP update methods. Prompted by the recent critiques of Hinton [2022], the forward-forward (FF) algorithm emerges as a more neurophysiologically aligned alternative, using dual forward passes with positive and negative data to simplify the training process and accommodate non-differentiable elements. The recently proposed cascaded forward algorithm (CaFo) [Zhao et al., 2023] attaches a class predictor to each layer, where only the layer-wise predictors are locally trained, with each neural block being randomly initialized and remaining fixed throughout.

Our work aims to push the frontier of the non-BP training algorithm for GNNs, which is a field still in its infancy. A remarkable recent work along the line is FORWARDGNN [Park et al., 2023] inspired by the forward-forward algorithm. FORWARDGNN avoids the constraints imposed by BP via an effective layer-wise local forward training. It trains GNNs using a single forward pass with the assistant of a data augmentation strategy. The augmented graph structure integrates virtual nodes linked only to labeled nodes, leaving the local topology of unlabeled nodes unchanged. The augmentation strategy makes it possible to operate without generating negative inputs. Despite of its advantages, FORWARDGNN still suffers from the greed-based training strategy, and thus results in inferiority in prediction performance in comparison with traditional BP algorithm.

In contrast, our method does not necessitate data augmentation for graph data; instead, it directly utilizes the discrepancy between predictions and actual ground truth to update each layer. Our approach directly outputs the prediction of the multi-class distribution, eliminating the need to calculate the goodness between unlabeled nodes and virtual nodes. As a result, our method offers convenience for multi-class prediction tasks and gains improvements in prediction performance.

## 3 Preliminaries

### 3.1 Problem Definition

An attributed relational graph of $n$ nodes can be represented by $G = (\mathcal{V}, \mathcal{E}, \mathbf{X})$, where $\mathcal{V} = \{v_1, v_2, \cdots, v_n\}$ represents the set of $n$ nodes, and $\mathcal{E} = \{e_{ij}\}$ signifies the set of edges. $\mathbf{X} = \{\mathbf{x}_1^\mathsf{T}; \mathbf{x}_2^\mathsf{T}; \cdots ; \mathbf{x}_n^\mathsf{T}\} \in \mathbb{R}^{n \times d}$ is the attribute set for all nodes, with $\mathbf{x}_i$ being the $d$-dimensional attribute vector for node $v_i$. The adjacency matrix $\mathbf{A} = \{a_{ij}\} \in \mathbb{R}^{n \times n}$ denotes the topological structure of graph $G$, where $a_{ij} > 0$ if there exists an edge $e_{ij}$ between nodes $v_i$ and $v_j$ and $a_{ij} = 0$ otherwise.

For semi-supervised node classification, the node set $\mathcal{V}$ can be split into a labeled node set $\mathcal{V}_L \subset \mathcal{V}$ with attributes $\mathbf{X}_L \subset \mathbf{X}$ and an unlabeled one $\mathcal{V}_U = \mathcal{V}/\mathcal{V}_L$ with attributes $\mathbf{X}_U = \mathbf{X}/\mathbf{X}_L$.[2] We assume that each node belongs to exactly one class, and denote $\mathbf{y}_L = \{y_i\}$ as the ground-truth labels of node set $\mathcal{V}_L$ where $y_i$ denotes the class label of node $v_i \in \mathcal{V}_L$.

The objective of semi-supervised node classification is to train a classifier using the graph and the known labels $\mathbf{y}_L$, and then apply this classifier to predict the labels for the unlabeled nodes $\mathbf{v}_U$. Define a classifier $f_\theta : (\tilde{\mathbf{y}}_L, \tilde{\mathbf{y}}_U) \leftarrow f_\theta(\mathbf{X}, \mathbf{A}, \mathbf{y}_L)$, where $\theta$ is the parameters of model. $\tilde{\mathbf{y}}_L$ and $\tilde{\mathbf{y}}_U$ are the predicted labels of nodes $\mathbf{v}_L$ and $\mathbf{v}_U$ respectively. Generally, the goal is to make the predicted labels $\tilde{\mathbf{y}}_L$ align as closely as possible with the actual ground-truth labels $\mathbf{y}_L$ in favor of: $\theta^* = \arg\min_\theta d(\tilde{\mathbf{y}}_L, \mathbf{y}_L) = \arg\min_\theta d(f_\theta(\mathbf{X}, \mathbf{A}, \mathbf{y}_L), \mathbf{y}_L)$, where $d(\cdot, \cdot)$ represents a measure of some type of distance between two sets of labels.

---

[2]For notation conciseness, we abuse the set notation and matrix notation interchangeably whenever appropriate. For example, $\mathbf{X}$ represents both a set of $n$ attributes and a matrix.

## 3.2 Direct Feedback Alignment

While BP relies on symmetric weights for error propagation to hidden layers, there is evidence suggesting that symmetrical weight distribution may not be crucial for learning. For example, the study of feedback alignment (FA) shows that learning can still occur when errors are back propagated using randomly fixed weights. Direct feedback alignment (DFA) advances in this direction by directly transmitting output errors to each hidden layer through fixed linear feedback links. Specifically, for an $L$ layer network, feedback matrices $\mathbf{B}^{(l)} \in \mathbb{R}^{n_L \times n_l}$ are employed to replace the derivatives $\partial \mathbf{x}^{(L)} / \partial \mathbf{x}^{(l)}$ of output neurons with respect to hidden neurons in the $l$-th layer. The approximate gradient $\delta \mathbf{W}^{(l)}$ for the weights of the $l$-th hidden layer is then computed as: $\delta \mathbf{W}^{(l)} = \frac{\partial \mathcal{L}}{\partial \mathbf{x}^{(L)}} \mathbf{B}^{(l+1)} \frac{\partial \mathbf{x}^{(l+1)}}{\partial \mathbf{W}^{(l)}}$, where $\mathbf{x}^{(l)}$ represents the latent representation of a sample at the $l$-th layer, and $\mathcal{L}$ the loss value. In DFA, the feedback matrices assigned to hidden layers are randomly selected and remain unchanged throughout the training process. The effectiveness of DFA hinges on the alignment between forward weights and feedback matrices, leading to a congruence between the estimated and the actual gradient. When the angle between these gradients remains below 90 degrees, the update direction points to a downward trajectory. DFA has been successfully implemented in popular deep learning architectures such as fully-connected neural network (FC) [Zhang et al., 2017] and convolutional neural network (CNN) [Li et al., 2021a]. However, extending DFA to graph neural networks remains unexplored.

## 4 Proposed DFA-GNN

As depicted in Fig. 1, our DFA-GNN aims to train GNN models in non-BP framework by extending the DFA algorithm. Different from the original DFA algorithm designed on FC for Euclidean data, two key issues should be solved when extending it to GNN for graph data: (1) the original random feedback operations need to be reformulated to handle the dependence between samples (nodes); and (2) high-quality pseudo errors for test samples are required as they are not isolated from the training procedure. To this end, in Sec. 4.1 we redesign the random feedback strategy specified for graph data, and in Sec. 4.2 we develop a novel pseudo error generator for semi-supervised graph learning tasks. Finally, we provide deep insight of our DFA-GCN about its convergence and optimization in Sec. 4.3.

### 4.1 Generalizing DFA to GNN

The training process of traditional BP for GNN is listed in Algo. 1 of Appx. A.1, and it uses both a forward propagation and a backward one in each epoch. Generally, different GNN models may differ in the operations of aggregation and combination, and we provide a typical implementation in Algo. 1. We take graph convolutional network (GCN) [Kipf and Welling, 2016], one of the most classic and successful GNN models, as a case study to integrate DFA for graph learning.

For illustrative purpose, we consider a three-layer GCN model with ReLU for hidden activation and sigmoid for output activation. The forward propagation process could be written as:

$$
\begin{aligned}
\textbf{Layer 1}: \mathbf{H}^{(0)} = \mathbf{S}\mathbf{X}^{(0)}, \quad & \mathbf{X}^{(1)} = \mathrm{relu}(\mathbf{H}^{(0)}\mathbf{W}^{(0)}), \\
\textbf{Layer 2}: \mathbf{H}^{(1)} = \mathbf{S}\mathbf{X}^{(1)}, \quad & \mathbf{X}^{(2)} = \mathrm{relu}(\mathbf{H}^{(1)}\mathbf{W}^{(1)}), \\
\textbf{Output layer}: \mathbf{H}^{(2)} = \mathbf{S}\mathbf{X}^{(2)}, \quad & \mathbf{X}^{(3)} = \mathbf{H}^{(2)}\mathbf{W}^{(2)}, \quad \tilde{\mathbf{Y}} = \mathrm{sigmoid}(\mathbf{X}^{(3)}),
\end{aligned}
\tag{1}
$$

where $\mathbf{S} = \tilde{\mathbf{D}}^{-\frac{1}{2}} \tilde{\mathbf{A}} \tilde{\mathbf{D}}^{-\frac{1}{2}}$, $\tilde{\mathbf{A}} = \mathbf{A} + \mathbf{I}$ is the adjacency matrix of graph $G$ after adding self loop. $\tilde{\mathbf{D}} = \mathrm{diag}(\tilde{\mathbf{A}})$ the diagonal matrix of $\tilde{\mathbf{A}}$, $\mathbf{W}^{(l-1)}$ a trainable weight matrix of the $l$-th layer and $\sigma$ a non-linear activation function. $\mathbf{X}^{(l)} \in \mathbb{R}^{n \times d}$ denotes the latent representation matrix of the $l$-th layer and $\mathbf{X}^{(0)} = \mathbf{X}$. If we choose sigmoid activation function in the output layer and a binary cross-entropy (BCE) loss function, the loss $J$ for a graph with $n$ nodes and the gradient $\mathbf{E}$ at the output layer are calculated as:

$$
J = -\frac{1}{N} \sum_{m,k} \mathbf{Y}_{m,k} \log \tilde{\mathbf{Y}}_{m,k} + (1 - \mathbf{Y}_{m,k}) \log(1 - \tilde{\mathbf{Y}}_{m,k}),
\tag{2}
$$

$$
\mathbf{E} = \delta \mathbf{X}^{(3)} = \frac{\partial J}{\partial \mathbf{X}^{(3)}} = \tilde{\mathbf{Y}} - \mathbf{Y},
\tag{3}
$$

where $\tilde{\mathbf{Y}}$ and $\mathbf{Y} \in \mathbb{R}^{n \times c}$ are respectively prediction and one-hot ground truth; $m$ and $k$ are respectively sample index and output unit. It is important to highlight that $\mathbf{E}$ represents the exact **error** between the prediction and the ground truth. For GCN, the gradients for hidden layers are calculated according to Algo. 1 as: $\delta \mathbf{X}^{(2)} = \mathbf{S}^{\mathrm{T}} \delta \mathbf{H}^{(2)} = \mathbf{S}^{\mathrm{T}} \mathbf{E} \mathbf{W}^{(2)\mathrm{T}}, \delta \mathbf{X}^{(1)} = \mathbf{S}^{\mathrm{T}} \delta \mathbf{H}^{(1)} = \mathbf{S}^{\mathrm{T}} \delta \mathbf{X}^{(2)} \mathbf{W}^{(1)\mathrm{T}}$.

As demonstrated by the work of FF [Lillicrap et al., 2016] and DFA [Nøkland, 2016], learning can be effective when errors are back propagated using randomly fixed weights. Similarly, we establish parallel direct feedback links for each layer. We approximate the update directions for the hidden layers as follows:

$$\delta \mathbf{X}^{(2)} = \mathbf{S}^{\mathrm{T}} \mathbf{E} \mathbf{B}^{(2)}, \quad \delta \mathbf{X}^{(1)} = \mathbf{S}^{\mathrm{T}} \delta \mathbf{H}^{(1)} = \mathbf{S}^{\mathrm{T}} \delta \mathbf{X}^{(2)} \mathbf{B}^{(1)}, \tag{4}$$

where $\mathbf{B}^{(i)}$ is a fixed random weight matrix with appropriate dimension. $\delta \mathbf{X}^{(1)}$ can be then written as:

$$\delta \mathbf{X}^{(1)} = \mathbf{S}^{\mathrm{T}} \delta \mathbf{X}^{(2)} \mathbf{B}^{(1)} = \mathbf{S}^{\mathrm{T}} (\mathbf{S}^{\mathrm{T}} \mathbf{E} \mathbf{B}^{(2)}) \mathbf{B}^{(1)} = (\mathbf{S}^{\mathrm{T}})^2 \mathbf{E} \mathbf{B}^{(1)}, \tag{5}$$

and the weight updates for all layers are calculated as:

$$\delta \mathbf{W}^{(0)} = \mathbf{H}^{(0)\mathrm{T}} \delta \mathbf{X}^{(1)}, \quad \delta \mathbf{W}^{(1)} = \mathbf{H}^{(1)\mathrm{T}} \delta \mathbf{X}^{(2)}, \quad \delta \mathbf{W}^{(2)} = \mathbf{H}^{(2)\mathrm{T}} \mathbf{E}. \tag{6}$$

The above derivations can be easily extended to GCNs with more layers, as done in our experiments.

## 4.2 Pseudo Error Generator by Spreading Residual Errors

In the computation of gradient (Eq. 6), the labels of all nodes are required, but not all nodes are labeled in the semi-supervised learning task. To address this issue, we introduce a mechanism to generate errors, assigning a pseudo error to each unlabeled node. The underlying principle is the expectation that errors in initial predictions are likely to propagate along the graph edges. That is, an error at a given node $v$ suggests a higher likelihood of similar errors to its the neighbor nodes. This concept of error propagation across the graph is supported by previous studies [Jia and Benson, 2020]. Our approach draws inspiration from the strategy of residual propagation used in node regression tasks, and more broadly, from the frameworks of generalized least squares and correlated error models [Shalizi, 2013].

In semi-supervised graph learning, the error matrix $\mathbf{E} = \{\mathbf{e}_1^{\mathrm{T}}; \mathbf{e}_2^{\mathrm{T}}; \cdots ; \mathbf{e}_n^{\mathrm{T}}\} \in \mathbb{R}^{n \times c}$ as described in Eq. 3 is modified as the residual on the training nodes, while being set to zero for all other nodes. This adjustment entails initializing $\mathbf{e}_i$ as a zero vector for all nodes $v_i \in \mathcal{V}_U$.

The residuals in the rows of $\mathbf{E}$ for the training nodes are zero only when the forward process achieves perfect predictions. We utilize the label spreading technique [Zhou et al., 2003] to smooth the error with the goal of optimizing the following objective:

$$\mathbf{Z}^* = \arg \min_{\mathbf{Z} \in \mathbb{R}^{n \times c}} tr(\mathbf{Z}^{\mathrm{T}}(\mathbf{I} - \mathbf{S})\mathbf{Z}) + \mu \|\mathbf{Z} - \mathbf{E}\|_F^2, \tag{7}$$

where $tr(\cdot)$ denotes the trace of a matrix. The first term enhances the smoothness of the error estimation throughout the graph, while the second term ensures that the final solution stays consistent with the initial error estimate $\mathbf{E}$. Following the optimization methodology in Zhou et al. [2003] and Huang et al. [2021], the solution to Eq. 7 can be obtained through iterative processing

$$\mathbf{Z}^{(t+1)} = (1 - \alpha)\mathbf{E} + \alpha \mathbf{S} \mathbf{Z}^{(t)}, \quad \alpha = \frac{1}{1 + \mu}, \quad \mathbf{Z}^{(0)} = \mathbf{E}. \tag{8}$$

This process represents the diffusion of error, and such propagation is demonstrably appropriate within the context of regression problems under a Gaussian assumption. Nevertheless, for classification tasks like ours, the smoothed error $\mathbf{Z}^*$ might not align with the correct scale. Typically, $\|\mathbf{Z}^{(t+1)}\|_2 \leq (1 - \alpha)\|\mathbf{E}\|_2 + \alpha \|\mathbf{S}\|_2 \|\mathbf{E}^{(t)}\|_2 = (1 - \alpha)\|\mathbf{E}\|_2 + \alpha \|\mathbf{Z}^{(t)}\|_2$. Starting with $\mathbf{Z}^{(0)} = \mathbf{E}$, we find that $\|\mathbf{Z}^{(t)}\|_2 \leq \|\mathbf{E}\|_2$, indicating a need to adjust the scale of residuals adaptively. The aim is to match the magnitude of error in $\mathbf{Z}^*$ to that in $\mathbf{E}$ as closely as possible. Given that we only have accurate error information for labeled nodes, we use the average error across these nodes to estimate the appropriate scale. Specifically, with $\mathbf{e}_i, \mathbf{z}_i^* \in \mathbb{R}^c$ representing the $i$-th row of $\mathbf{E}$ and $\mathbf{Z}^*$ respectively, the adjusted error for an unlabeled node $j$ is calculated as: $\hat{\mathbf{e}}_j = \eta / \|\mathbf{z}_j^*\|_1 \cdot \mathbf{z}_j^*$, in which $\eta = \frac{1}{|\mathcal{V}_L|} \sum_{v_i \in \mathcal{V}_L} \|\mathbf{e}_i\|_1$.

Give the rescaled pseudo error $\hat{\mathbf{e}}$ for each unlabeled node, we define $\hat{\mathbf{E}} \in \mathbb{R}^{n \times c}$, where the $i$-th row is set to $\mathbf{e}_i^{\mathrm{T}}$ for nodes $v_i \in \mathcal{V}_L$ and to $\hat{\mathbf{e}}_i^{\mathrm{T}}$ for other nodes. The matrix $\hat{\mathbf{E}}$ can then be directly utilized in Eqs. 5 and 6 for GCN training. Nevertheless, not all rescaled error of unlabeled nodes are accurate and useful. To address this, a mask is implemented to filter out these nodes. Define $\hat{\mathbf{Y}} = \tilde{\mathbf{Y}} - \hat{\mathbf{E}}$ as the corrected prediction. For labeled nodes, this corrected prediction equals to the one-hot ground truth. We introduce a mask vector $\mathbf{p} \in \mathbb{R}^n$ to facilitate this process. Formally, the setup is as follows:

$$p_i = \begin{cases} 1, & \text{if } \mathrm{count}(\hat{\mathbf{y}}_i > \epsilon) = 1, \\ 0, & \text{otherwise}, \end{cases} \tag{9}$$

where $p_i$ is the $i$-th element ($1 \leq i \leq n$) of the vector $\mathbf{p}$, $\epsilon$ a manually set threshold for controlling filtering and $\mathrm{count}(\cdot)$ a counting function. Eq. 9 focuses on retaining only those well predicted nodes, characterized by a single category being identified as positive and the rest as negative. With the mask, the filtering operations could be performed on $\hat{\mathbf{E}}$ and $\mathbf{S}$ by row through the mask. The weight updates in Eq. 6 are modified as:

$$\delta\mathbf{W}^{(0)} = \mathbf{H}^{(0)\mathrm{T}}\mathbf{S}_f^{(2)\mathrm{T}}\hat{\mathbf{E}}_f\mathbf{B}^{(1)}, \quad \delta\mathbf{W}^{(1)} = \mathbf{H}^{(1)\mathrm{T}}\mathbf{S}_f^{(1)\mathrm{T}}\hat{\mathbf{E}}_f\mathbf{B}^{(2)}, \quad \delta\mathbf{W}^{(2)} = \mathbf{H}_f^{(2)\mathrm{T}}\hat{\mathbf{E}}_f, \tag{10}$$

where $\mathbf{S}_f^{(k)}$, $\hat{\mathbf{E}}_f$, $\mathbf{H}_f^{(k)}$ represent the row filtering of $\mathbf{S}^k$, $\hat{\mathbf{E}}$ and $\mathbf{H}^{(k)}$ respectively, according to the mask $\mathbf{p}$. As $(\mathbf{S}^{\mathrm{T}})^k\hat{\mathbf{E}}$ exactly denotes the accumulation of errors related to the $k$-hop neighbors of each node, the difference between Eq. 6 and Eq. 10 lies in the partial sampling of neighbors to approximate the update directions, as compared with using all neighbors. Given that only poorly predicted nodes are excluded and they constitute a small fraction, the update directions in both approaches generally remain aligned. The formal description of our algorithm is provided in Appx. A.2.

## 4.3  Insights of DFA-GNN

DFA-GNN provides a non-BP training procedure for GNN. For GCN we provide a detailed analysis on how such an asymmetric feedback path introduced in Sec. 4.1 can provide learning by aligning the gradients of backward propagation and forward propagation with its own. Nøkland [2016] has originally proved the conclusion for fully-connected layer architectures. We can show that this conclusion is equally valid for the GCN architecture, and provide the detailed proof in Appx. A.3.

Experiments in Fig. 2(a) validate the dynamic process of alignment between $\mathbf{B}$ and $\mathbf{W}$ during training, and the weight alignment leads to gradient alignment because for weight alignment of DFA:

$$\mathbf{W}^{(0<l<L)} \propto \mathbf{B}^{(l)\mathrm{T}}\mathbf{B}^{(l+1)}, \quad \mathbf{W}^{(L)} \propto \mathbf{B}^{(L)\mathrm{T}}, \tag{11}$$

where the symbol $\propto$ represents a positive scalar multiple relationship. As gradient alignment requires $\delta\mathbf{X}_{\mathrm{DFA}}^{(l)} \propto \delta\mathbf{X}_{\mathrm{BP}}^{(l)}$, i.e., $(\mathbf{S}^{\mathrm{T}})^{L-l}\mathbf{E}\mathbf{B}^{(l)} \propto \mathbf{S}^{\mathrm{T}}\delta\mathbf{X}^{(l+1)}\mathbf{W}^{(l)\mathrm{T}}$, the weight alignment directly implies gradient alignment if the feedback matrices are assumed right-orthogonal, i.e., $\mathbf{B}\mathbf{B}^{\mathrm{T}} = \mathbf{I}$. This assumption holds if the feedback matrices elements are sampled i.i.d. from a Gaussian distribution since $\mathbb{E}[\mathbf{B}\mathbf{B}^{\mathrm{T}}] \propto \mathbf{I}$, hence Eq. 11 induces the weights, by the orthogonality condition, to cancel out by pairs of two:

$$\mathbf{S}^{\mathrm{T}}\delta\mathbf{X}^{(l+1)}\mathbf{W}^{(l)\mathrm{T}} \propto (\mathbf{S}^{\mathrm{T}})^{L-l}\mathbf{E}\mathbf{B}^{(L)}\mathbf{B}^{(L)\mathrm{T}}\cdots\mathbf{B}^{(l+1)\mathrm{T}}\mathbf{B}^{(l)} = (\mathbf{S}^{\mathrm{T}})^{L-l}\mathbf{E}\mathbf{B}^{(l)}. \tag{12}$$

The alignments of weights and gradients make our method trained with DFA tend to converge to a specific region within the landscape, guided by the structure of the feedback matrices, while the optimization paths trained with BP according to stochastic gradient descent often exhibit divergent within the loss landscape, as shown in Fig. 2 (d).

For deeper understanding of the training mechanism in our method, we divide the entire training process into three stages: Stage 1 (train layers 1 and 2 while freezing layer 3), Stage 2 (freeze layers 1 and 2 while training layer 3) and Stage 3 (train layers 1 and 2 while freezing layer 3). The results in Figs. 2(b,c) show a strong correlation between weight alignment and the fitting degree of the model, as indicated by the loss. Notably, even though our method updates parameters of each layer in parallel, the effective update follows a backward-to-forward manner. As shown in Fig. 2 (b), when the parameters of layer $l$ are not effectively learned, updating the preceding layers does not enhance fitting ability of the model. This behavior contrasts with the characteristics of traditional BP, which indicates that the alignment of weights and gradients also adhere to a backward-to-forward sequence.

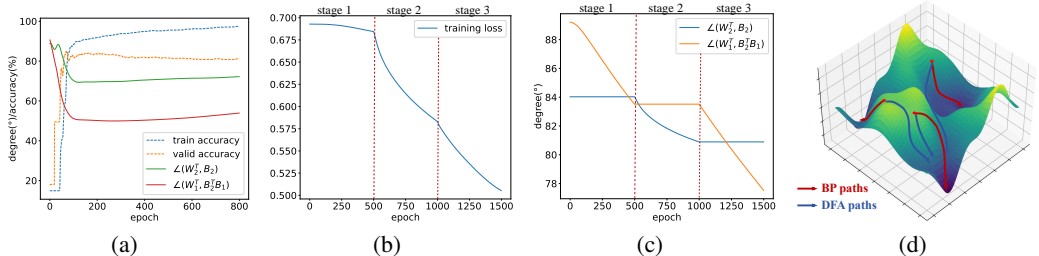

Figure 2: For a three-layer GCN model trained by DFA-GNN on Cora, (a) the accuracy and angle between $\mathbf{W}$ and $\mathbf{B}$; (b) the change in loss across different stages; (c) the change in angle between $\mathbf{W}$ and $\mathbf{B}$ across different stages; (d) difference in optimization direction between BP and our method.

Table 1: Results on datasets: mean accuracy (%) $\pm$ 95% confidence interval. The best result on each dataset is indicated with **bold**.

|  | BP | PEPITA | CaFo+MSE | CaFo+CE | FF+LA | FF+VN | SF | Ours |
|---|---|---|---|---|---|---|---|---|
| Cora | 86.04 | 25.78 | 71.79 | 71.78 | 84.20 | 74.50 | 84.54 | **87.72** |
|  | ±0.62 | ±6.24 | ±1.76 | ±1.71 | ±0.85 | ±1.54 | ±0.77 | ±1.63 |
| CiteSeer | 78.20 | 21.24 | 65.43 | 63.12 | 75.25 | 69.97 | 73.84 | **80.49** |
|  | ±0.57 | ±2.63 | ±0.99 | ±1.15 | ±1.09 | ±1.08 | ±1.02 | ±0.41 |
| PubMeb | 85.24 | 36.13 | 77.66 | 78.29 | 83.68 | 79.60 | 84.68 | **86.28** |
|  | ±0.28 | ±10.88 | ±0.82 | ±0.64 | ±0.38 | ±0.62 | ±0.61 | ±0.67 |
| Photo | 93.03 | 70.63 | 89.48 | 90.59 | 86.39 | 15.56 | 92.48 | **93.04** |
|  | ±0.59 | ±7.13 | ±0.33 | ±0.26 | ±3.46 | ±9.38 | ±0.33 | ±0.31 |
| Computer | **89.48** | 63.25 | 83.02 | 82.94 | 75.87 | 12.27 | 84.04 | 86.72 |
|  | ±0.37 | ±8.78 | ±0.59 | ±0.73 | ±4.55 | ±1.60 | ±0.65 | ±0.68 |
| Texas | 54.26 | 39.67 | 56.39 | 31.14 | 59.67 | 19.67 | 37.71 | **79.51** |
|  | ±3.44 | ±18.02 | ±3.44 | ±3.44 | ±3.28 | ±12.30 | ±2.95 | ±1.97 |
| Cornell | 71.31 | 50.49 | 28.85 | 36.72 | 45.90 | 18.52 | 60.66 | **75.24** |
|  | ±4.43 | ±20.66 | ±2.95 | ±5.73 | ±8.04 | ±12.46 | ±5.25 | ±4.92 |
| Actor | 31.94 | 21.55 | 23.83 | 23.83 | 33.58 | 15.05 | 28.33 | **34.07** |
|  | ±0.88 | ±3.86 | ±0.76 | ±0.64 | ±1.54 | ±7.26 | ±1.38 | ±0.75 |
| Chameleon | 41.28 | 36.97 | 37.36 | 36.48 | 33.21 | 24.76 | **42.35** | 41.19 |
|  | ±2.29 | ±2.45 | ±1.91 | ±1.44 | ±1.99 | ±2.61 | ±2.27 | ±1.56 |
| Squirrel | 37.81 | 33.99 | 31.00 | 31.00 | 33.66 | 18.91 | 36.00 | **38.17** |
|  | ±0.71 | ±1.24 | ±1.18 | ±0.92 | ±0.87 | ±4.28 | ±1.50 | ±2.21 |

## 5 Experiments

### 5.1 Comparison with Baseline Training Algorithms

We evaluate our method on 10 benchmark datasets across various domains and compare it with the BP [Rumelhart et al., 1986], PEPITA [Dellaferrera and Kreiman, 2022], two versions of the FF (*abbr.* FF+LA, FF+VN) [Hinton, 2022, Park et al., 2023], two versions of the CaFo (*abbr.* CaFo+MSE, CaFo+CE) [Zhao et al., 2023] and the FORWARDGNN-SF (*abbr.* SF) [Park et al., 2023]. Detailed datasets and experimental setup information can be found in Appx. A.4 and Appx. A.5, respectively. The comparative analysis of various algorithms on benchmark datasets is summarized in Tab. 1. While the non-BP methods such as PEPITA, CaFo and FF have proven effective for architectures involving fully-connected and convolutional layers with Euclidean data, they exhibit weaker performance with non-Euclidean graph data. This is primarily due to the unique challenges posed by graph data.

Firstly, in typical fully-connected and convolutional layers, shallow layers capture coarse-grained features while deep layers handle fine-grained features, with these two types of features usually being highly correlated. However, in GNNs, different layers aggregate information from varying neighborhood ranges, resulting in layers that often contain uncorrelated information. Particularly in heterophilic graphs, the information extracted by deep and shallow layers may be entirely unrelated

Table 2: Ablation study results on different datasets with proposed designs.

| EG | NF | Cora | CiteSeer | PubMed | Actor | Chameleon | Squirrel |
|----|----|------|----------|--------|-------|-----------|----------|
| ✗ | ✗ | 83.02±1.36 | 78.17±0.71 | 82.92±0.42 | 30.72±1.72 | 39.09±1.17 | 34.61±1.01 |
| ✓ | ✗ | 86.70±1.00 | 79.26±0.90 | 84.01±0.23 | 31.97±1.46 | 39.62±1.31 | 34.87±1.78 |
| ✓ | ✓ | **87.72**±1.63 | **80.49**±0.41 | **86.28**±0.67 | **34.07**±0.75 | **41.19**±1.56 | **38.17**±2.21 |

or even have opposing effects on predictions. This lack of correlation complicates the application of layer-wise optimization strategies, which rely on greedy strategies and local loss calculations common in traditional networks. This is a key reason for the underperformance of methods like PEPITA, CaFo and FF in GNNs, as evidenced in Tab. 1, particularly on datasets characterized by low homophily. Secondly, since graph data do not adhere to the *i.i.d.* assumption, sampling positive and negative samples based on features (FF+LA) and topology (FF+VN) for the FF algorithm can be unreliable, potentially leading to inconsistent results. Notably, FF+VN modifies the original graph topology by introducing virtual nodes into both positive and negative graphs, which results in overall unsatisfactory performance in benchmarks. For CaFo, the rigidity in fixing the parameters of each block, with only the predictors being learnable, further constrains its adaptability. As the most recently proposed non-BP GNN training approach, SF shows a performance that is superior to PEPITA, CaFo and FF but still lags behind the traditional BP method on most datasets. The reason lies in that SF also introduces virtual nodes that disrupt graph topology and employs a layer-wise training strategy. By contrast, our method well adapts to graph data and gains significant improvement in testing accuracy in comparison with the baseline algorithms, achieving the best or second-best results across all datasets.

The training times for each method are shown in Appx. A.6. Our approach demonstrates a general time advantage over CaFo, FF and SF. Our method includes a forward propagation and a parameter update where all layers execute in parallel during each iteration, which offers greater parallelism compared with layer-wise update methods like CaFo, FF and SF. Our method has a higher training time consumption compared with BP, primarily due to the additional time needed for generating pseudo errors and filtering masks, as discussed in Sec. 4.2.

## 5.2 Ablation Study

We ablate the proposed method to study the importance of designs in DFA-GNN. Two designs are studied including the pseudo error generator (*abbr.* EG) and the node filter (*abbr.* NF). For trials with EG, the pseudo error generator is applied according to Eq. 7 to assign a pseudo error for each unlabeled node. It worth noting that when ER is removed from the method, only the errors of labeled nodes are used for the updating of parameters according to Eq. 6. For trials with NF, the mask calculated as Eq. 9 is introduced in training process and the parameters are updated according to Eq. 10. The ablation results are included in Tab. 2. We note that even the most naive version of DFA-GNN elaborated in Sec. 4.1 achieves comparable results in comparison with BP. Furthermore, both the two designs introduced in Sec. 4.2 contribute to our training framework, making significant enhancement to DFA-GNN to outperform BP method.

## 5.3 Visualization of Convergence

To better illustrate the training convergence of DFA-GNN, we plot the training and validation accuracy of the proposed DFA-GNN and BP over training epochs on three datasets, as shown in Fig. 3. In general, our method shows similar convergence to BP. For both BP and our method, the convergence of validation accuracy occurs much earlier than that of training accuracy due to overfitting. The validation convergent epoch of DFA-GNN is nearly the same as BP on Cora and CiteSeer (around 100 epochs), while it is 100 epochs later than BP on PubMed (around 200 epochs). Our method achieves better validation accuracy on all these datasets and suffers less from overfitting compared with BP. In terms of training accuracy, the convergence of our method is slightly slower than BP because the update direction of our method is not exactly opposite to the gradient direction but maintains a small angle. Since our method considers both the errors of labeled nodes and pseudo errors of unlabeled nodes as supervision information, which is different from BP that only uses the loss of labeled nodes

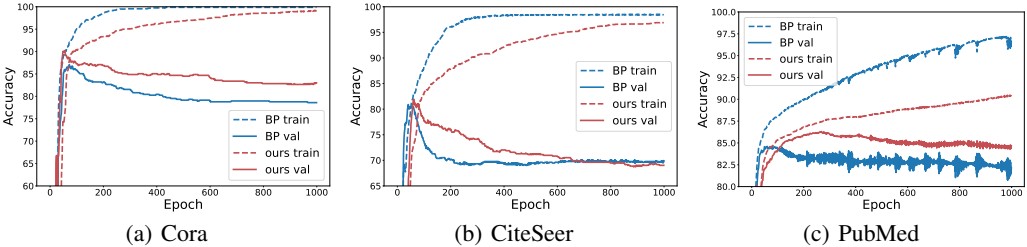

|  |  |  |
|---|---|---|
| (a) Cora | (b) CiteSeer | (c) PubMed |

Figure 3: Visualization of the convergence of BP and our method on Cora, CiteSeer, and PubMed.

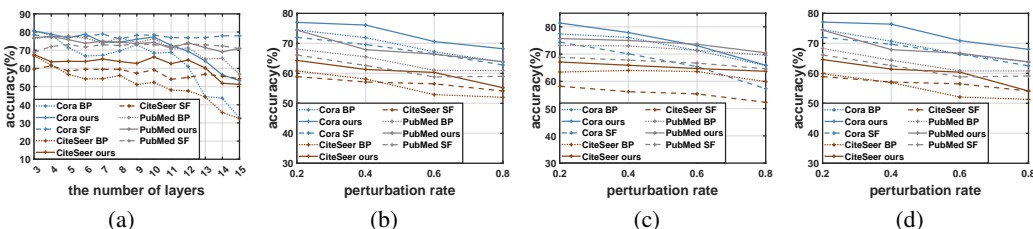

|  |  |  |  |
|---|---|---|---|
| (a) | (b) | (c) | (d) |

Figure 4: (a) Test accuracy with the model layers increasing. (b-d) Test accuracy with the perturbation rate (b: *add*, c: *remove*, d: *flip*) increasing.

for supervision, the convergent value of training accuracy for our method is slightly lower than BP. However, this does not affect our method achieving better validation results.

### 5.4 Robustness Analysis

We focus on over-smoothing and random structural attack, two common sources of perturbation that reduce GNN performance. Over-smoothing [Keriven, 2022] is a problematic issue in GNNs, stemming from the aggregation mechanism within GNNs, which hinders the expansion of GNN models to a large number of layers. We test the robustness of our method against over-smoothing in Fig. 4 (a). Our method demonstrates greater robustness compared with BP, particularly when dealing with architectures that have a large number of layers. This enhanced robustness is due to the fact that the global loss directly contributes to the optimization of each individual layer. SF is less effected by over-smoothing due to its layer-wise optimization with local loss. However, its performance largely depends on shallow layers and the best performance on each dataset is inferior to ours.

For random structural attack [Li et al., 2021b], three random attack types are implemented on the original graph topology with a perturbation rate $\lambda$ from 0.2 to 0.8. To better compare the robustness of different methods, we employ a more challenging experimental setup with a sparse supervision pattern, where each class has only 20 labeled nodes. The node split follows Kipf and Welling [2016]. The detailed operating description for attacking type is summarized as follows: (1) *add*: randomly add $\lambda|\mathcal{E}|$ edges to the original graph for a denser topology; (2) *remove*: randomly remove $\lambda|\mathcal{E}|$ edges from the original graph for a sparser topology; and (3) *flip*: randomly choose $\lambda|\mathcal{E}|$ node pairs, and remove the edge if there exists one between the pair, and otherwise add an edge to connect the pair.

Our method is less sensitive to all types of perturbations as shown in Figs. 4 (b-d), consistently outperforms two approaches on each trial, and exhibits exciting robustness even under a high perturbation rate. As the pseudo error generator derives pseudo error for each unlabeled node to update each layer, this supervision generation mechanism helps enhance robustness of the model against noise and attacks. Interestingly, a comparison of results across three different types of attacks shows that removing edges has the least adverse effect, suggesting that injecting incorrect topological information could be more detrimental to GNN performance than losing valuable original topology.

### 5.5 Scalability on Large Datasets

Our method is well-suited for large datasets. When the graph scale is large, we can use edge indices instead of an adjacency matrix to store the graph. For forward propagation (Eq. 1), the complexity

Table 3: Results on three large datasets. *OOM* denotes out of memory.

|           | FF+LA | FF+VN | PEPITA | CaFo+MSE | CaFo+CE | SF    | BP      | Ours    |
|-----------|-------|-------|--------|----------|---------|-------|---------|---------|
| Flickr    | 6.09  | 42.40 | 49.28  | 50.02    | 49.69   | 46.47 | **50.79** | 49.80   |
| Reddit    | 12.44 | *OOM* | *OOM*  | 88.15    | 91.55   | 94.38 | 94.34   | **94.49** |
| ogbn-arxiv| 56.38 | 19.84 | 35.16  | 53.51    | 60.57   | 66.54 | **68.78** | 67.83   |

Table 4: Performance of our method integrated with different GNN models.

|              | Cora        | CiteSeer    | PubMeb      | Photo       | Computer    |
|--------------|-------------|-------------|-------------|-------------|-------------|
| SGC+BP       | 85.30±0.79  | 79.45±0.87  | 79.72±0.35  | 91.71±0.44  | 85.07±0.27  |
| SGC+ours     | **88.60**±0.85 | **81.02**±0.78 | **80.40**±0.18 | **92.01**±0.29 | **85.94**±0.31 |
| GAT+BP       | 85.35±0.62  | 79.09±1.31  | 85.78±0.34  | 93.16±0.43  | **88.91**+0.78 |
| GAT+ours     | **86.96**±1.08 | **79.77**±1.15 | **86.18**±0.32 | **93.24**±0.51 | 87.38±0.66  |
| GraphSage+BP | 87.04±0.84  | 79.65±1.11  | **88.32**±0.28 | 92.89±0.51  | 88.00±0.29  |
| GraphSage+ours | **87.88**±1.00 | **79.69**±1.33 | 87.65±0.28  | **93.95**±0.33 | **88.14**±0.39 |
| APPNP+BP     | 85.75±0.89  | **80.46**±0.37 | **85.81**±0.28 | 91.55±0.81  | 85.50±0.57  |
| APPNP+ours   | **85.81**±1.11 | 79.47±0.80  | 85.75±0.32  | **91.66**±0.39 | **85.68**±0.43 |
| ChebNet+BP   | 83.45±1.07  | 76.93±0.71  | **87.09**±0.31 | 90.89±0.74  | 86.51±0.78  |
| ChebNet+ours | **85.53**±1.36 | **77.85**±0.96 | 86.43±0.37  | **92.65**±0.49 | **87.02**±0.62 |

of neighbor aggregation can be reduced from $\mathcal{O}(n^2 d)$ to $\mathcal{O}(|\mathcal{E}|d)$, where $|\mathcal{E}|$ denotes the number of edges. For direct feedback alignment (Eq. 10), as $(\mathbf{S}^{\mathrm{T}})^k \hat{\mathbf{E}}$ is exactly the aggregation of errors for $k$ times, the time and space complexity can be reduced to $\mathcal{O}(kc|\mathcal{E}|)$, without the need to calculate the $k$-th power of the adjacency matrix. Similarly, complexity reduction can also be achieved in the node filtering process. The experimental results on the Flickr [Zeng et al., 2020], Reddit [Hamilton et al., 2017], and ogbn-arxiv [Hu et al., 2020] datasets, as presented in Tab. 3, demonstrate that our method is effective on large-scale datasets, delivering strong performance. Our method achieves results comparable to BP while surpassing other non-BP methods, all with a small memory footprint (2043 MiB for Flickr, 11675 MiB for Reddit, and 2277 MiB for ogbn-arxiv). This indicates that our method not only scales well to large graphs but also maintains efficiency in terms of space usage.

### 5.6 Portability Analysis

We apply our training algorithm to five popular GNN models [Wu et al., 2019, Veličković et al., 2018, Hamilton et al., 2017, Gasteiger et al., 2018, Defferrard et al., 2016] and report the mean accuracy across ten random splits in Tab. 4. Each of the testing models is modified to fit our framework. Specifically, for SGC, which only has a single learnable linear output layer, the training of our framework involves no direct feedback but only incorporates the pseudo error generator and node filter. For GraphSage, we utilize a mean-aggregator for message aggregation. Our observations indicate that our method can be effectively ported to mainstream GNN models. All test models integrated with our algorithm work well and surpass the performance of traditional BP in most scenarios. It demonstrates the effectiveness of our method across various GNN models and underscores its excellent portability and potential generalization ability to other innovative GNN models.

## 6 Conclusion

In this paper, we investigate the potential of non-backpropagation training methods within the context of graph learning. We adapt the direct feedback alignment algorithm for training graph neural networks on graph data and introduce DFA-GNN. This new approach incorporates a meticulously designed random feedback strategy, a pseudo error generator, and a node filter to effectively spread residual errors. Through mathematical formulations, we demonstrate that our method can align with backpropagation in terms of parameter update gradients and perform effective training. Extensive experiments on real-world datasets confirm the effectiveness, efficiency, robustness and versatility of our proposed forward graph learning framework.

## Acknowledgments

This work is supported by the National Nature Science Foundation of China (Nos. 62076021 and 62376020). Haibin Ling was not supported by any fund for this work.

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

# A APPENDIX

Here we provide some implementation details of our methods to help readers further understand the algorithms and experiments in this paper.

## A.1 GNN Training Process of BP

---
**Algorithm 1** GNN Training Process of BP
---
**Input**: $G$, $\mathbf{X}$, $\mathbf{W}$, $\mathbf{y}_L$, *max_epoch*.
**Output**: $\tilde{\mathbf{y}}$, $\mathbf{W}$.

1: **for** *epoch* in [0, 1, ..., *max_epoch*-1] **do**
2:    **for** $l$ in [0, 1, ..., $L-1$] **do** {Forward propagation}
3:       **for** $v$ in $\mathbf{v}$ **do**
4:          $\mathbf{h}_v^{(l)} = \sum_{u \in \tilde{N}(v)} e_{vu} \cdot \mathbf{x}_u^{(l)}$; {*Aggregation*}
5:       **end for**
6:       $\mathbf{X}^{(l+1)} = \sigma(\mathbf{H}^{(l)}\mathbf{W}^{(l)})$; {*Combination*}
7:    **end for**
8:    $loss = Loss\_computing(\mathbf{X}^{(L)}, \mathbf{y}_L)$
9:    **for** $l$ in [$L-1$, $L-2$, ..., 0] **do** {Backward propagation}
10:      $\delta\mathbf{W}^{(l)} = \mathbf{H}^{(l)T}\delta\mathbf{X}^{(l+1)}$；{*Combination*}
11:      $\delta\mathbf{H}^{(l)} = \delta\mathbf{X}^{(l+1)}\mathbf{W}^{(l)T}$；{*Combination*}
12:      **for** $v$ in $\mathbf{v}$ **do**
13:         $\delta\mathbf{x}_v^{(l)} = \sum_{u \in \tilde{N}(v)} e_{uv} \cdot \delta\mathbf{h}_u^{(l)}$; {*Aggregation*}
14:      **end for**
15:    **end for**
16:    $Weight\_updating(epoch, \mathbf{W}^{(0)}, \delta\mathbf{W}^{(0)}, ..., \mathbf{W}^{(L-1)}, \delta\mathbf{W}^{(L-1)})$;
17: **end for**
18: $\tilde{\mathbf{y}} = \arg\max \tilde{\mathbf{Y}} = \arg\max \mathbf{X}^{(L)}$.
---

## A.2 GCN Training Process of Our Method

---
**Algorithm 2** GCN Training Process
---
**Input**: $G$, $\mathbf{X}$, $\mathbf{W}$, $\mathbf{y}_L$, *max_epoch*.
**Output**: $\tilde{\mathbf{y}}$, $\mathbf{W}$.

1:  Initialize $\{\mathbf{B}^{(1)}, \mathbf{B}^{(2)}, ..., \mathbf{B}^{(L-1)}\}$;
2: **for** *epoch* in [0, 1, ..., *max_epoch*-1] **do**
3:    Compute the prediction $\tilde{\mathbf{Y}}$ given by Eq. 1; {Forward propagation}
4:    Compute the error $\mathbf{E}$ using $\tilde{\mathbf{Y}}$, $\mathbf{y}_L$ for labeled nodes according to Eq. 3;
5:    Compute pseudo error $\mathbf{Z}^*$ for unlabeled nodes by Eqs. 8, and rescale it to get rescaled error $\hat{\mathbf{E}}$;
6:    Compute mask vector $\mathbf{p}$ given by Eq. 9;
7:    **for** $l$ in [0, 1, ..., $L-1$] **do**
8:      **if** $l == L-1$ **then**
9:         Compute $\delta\mathbf{W}^{(l)}$ using $\hat{\mathbf{E}}$, $\mathbf{p}$ according to Eq. 10; {Direct feedback}
10:      **else**
11:         Compute $\delta\mathbf{W}^{(l)}$ using $\hat{\mathbf{E}}$, $\mathbf{p}$ and $\mathbf{B}^{(l+1)}$ according to Eq. 10; {Direct feedback}
12:      **end if**
13:    **end for**
14:    $Weight\_updating(epoch, \mathbf{W}^{(0)}, \delta\mathbf{W}^{(0)}, ..., \mathbf{W}^{(L-1)}, \delta\mathbf{W}^{(L-1)})$;
15: **end for**
16: $\tilde{\mathbf{y}} = \arg\max \tilde{\mathbf{Y}}$.
---

## A.3 Proof of Theorem in Sec. 4.3

**Theorem.** For a GCN model with two hidden layers $k$ and $k+1$ where $k$ connects to $k+1$ in sequence, we have $\mathbf{x}^{(k+1)} = \sigma(\mathbf{a}^{(k+1)})$ and $\mathbf{a}^{(k+1)} = g(\mathbf{W}\mathbf{x}^{(k)})$, where $\sigma$ is the activation function and $g(\cdot)$ the aggregation operation in Algo. 1. Let the layers be updated according to the non-zero update directions $\delta\mathbf{x}^{(k)}$ and $\delta\mathbf{x}^{(k+1)}$ where $\frac{\delta\mathbf{x}^{(k)}}{\|\delta\mathbf{x}^{(k)}\|}$ and $\frac{\delta\mathbf{x}^{(k+1)}}{\|\delta\mathbf{x}^{(k+1)}\|}$ are constant for each data point. The negative update directions will minimize the following layer-wise criterion:

$$\mathbf{P} = \mathbf{P}^{(k)} + \mathbf{P}^{(k+1)} = \frac{\delta\mathbf{x}^{(k)\mathrm{T}}\mathbf{x}^{(k)}}{\|\delta\mathbf{x}^{(k)}\|} + \frac{\delta\mathbf{x}^{(k+1)\mathrm{T}}\mathbf{x}^{(k+1)}}{\|\delta\mathbf{x}^{(k+1)}\|}. \tag{13}$$

Minimizing $\mathbf{P}$ will lead to an increase in the gradient, thereby enhancing the alignment criterion:

$$\mathbf{Q} = \mathbf{Q}^{(k)} + \mathbf{Q}^{(k+1)} = \frac{\delta\mathbf{x}^{(k)\mathrm{T}}\mathbf{c}^{(k)}}{\|\delta\mathbf{x}^{(k)}\|} + \frac{\delta\mathbf{x}^{(k+1)\mathrm{T}}\mathbf{c}^{(k+1)}}{\|\delta\mathbf{x}^{(k+1)}\|}, \tag{14}$$

where

$$\mathbf{c}^{(k)} = \frac{\partial\mathbf{x}^{(k+1)}}{\partial\mathbf{x}^{(k)}}\delta\mathbf{x}^{(k+1)} = \mathbf{W}^{\mathrm{T}}g'(\delta\mathbf{x}^{(k+1)} \odot \sigma'(\mathbf{a}^{(k+1)})),$$

$$\mathbf{c}^{(k+1)} = \frac{\partial\mathbf{x}^{(k+1)}}{\partial\mathbf{x}^{(k)\mathrm{T}}}\delta\mathbf{x}^{(k)} = g'(\mathbf{W}\delta\mathbf{x}^{(k)}) \odot \sigma'(\mathbf{a}^{(k+1)}). \tag{15}$$

$g'(\cdot)$ is the aggregation of gradients in Algo. 1, (line 13). If $\mathbf{Q}^{(k)} > 0$, then $-\delta\mathbf{x}^{(k)}$ serves as a direction of descent to minimize $\mathbf{P}^{(k+1)}$.

**Proof.** Let $i$ be the any of the layers $k$ or $k+1$, and prescribed update $-\delta\mathbf{x}^{(i)}$ is the steepest descent direction to minimize $\mathbf{P}^{(i)}$. Since any partial derivative of $\frac{\delta\mathbf{x}^{(i)}}{\|\delta\mathbf{x}^{(i)}\|}$ is zero, we have:

$$-\frac{\partial\mathbf{P}^{(i)}}{\partial\mathbf{x}^{(i)}} = -\frac{\partial}{\partial\mathbf{x}^{(i)}}\Big[\frac{\delta\mathbf{x}^{(i)\mathrm{T}}\mathbf{x}^{(i)}}{\|\delta\mathbf{x}^{(i)}\|}\Big] = -\frac{\partial}{\partial\mathbf{x}^{(i)}}\Big[\frac{\delta\mathbf{x}^{(i)}}{\|\delta\mathbf{x}^{(i)}\|}\Big]\mathbf{x}^{(i)} - \frac{\partial\mathbf{x}^{(i)}}{\partial\mathbf{x}^{(i)}}\frac{\delta\mathbf{x}^{(i)}}{\|\delta\mathbf{x}^{(i)}\|} = -\alpha^{(i)}\delta\mathbf{x}^{(i)}, \tag{16}$$

where $\alpha^{(i)} = \frac{1}{\|\delta\mathbf{x}^{(i)}\|} > 0$. As $\delta\mathbf{a}^{(i)} = \frac{\partial\mathbf{x}^{(i)}}{\partial\mathbf{a}^{(i)}}\delta\mathbf{x}^{(i)} = \delta\mathbf{x}^{(i)} \odot \sigma'(\mathbf{a}^{(i)})$, the gradients maximizing $\mathbf{Q}^{(k)}$ and $\mathbf{Q}^{(k+1)}$ are:

$$\frac{\partial\mathbf{Q}^{(i)}}{\partial\mathbf{c}^{(i)}} = \frac{\partial}{\partial\mathbf{c}^{(i)}}\Big[\frac{\delta\mathbf{x}^{(i)\mathrm{T}}\mathbf{c}^{(i)}}{\|\delta\mathbf{x}^{(i)}\|}\Big] = \frac{\partial}{\partial\mathbf{c}^{(i)}}\Big[\frac{\delta\mathbf{x}^{(i)}}{\|\delta\mathbf{x}^{(i)}\|}\Big]\mathbf{c}^{(i)} + \frac{\partial\mathbf{c}^{(i)}}{\partial\mathbf{c}^{(i)}}\frac{\delta\mathbf{x}^{(i)}}{\|\delta\mathbf{x}^{(i)}\|} = \alpha^{(i)}\delta\mathbf{x}^{(i)},$$

$$\frac{\partial\mathbf{Q}^{(k+1)}}{\partial\mathbf{W}} = \frac{\partial\mathbf{Q}^{(k+1)}}{\partial\mathbf{c}^{(k+1)}}\frac{\partial\mathbf{c}^{(k+1)}}{\partial\mathbf{W}} = \alpha^{(k+1)}g'(\delta\mathbf{x}^{(k+1)} \odot \sigma'(\mathbf{a}^{(k+1)}))\delta\mathbf{x}^{(k)\mathrm{T}} = \alpha^{(k+1)}g'(\delta\mathbf{a}^{(k+1)})\delta\mathbf{x}^{(k)\mathrm{T}},$$

$$\frac{\partial\mathbf{Q}^{(k)}}{\partial\mathbf{W}} = \frac{\partial\mathbf{c}^{(k)}}{\partial\mathbf{W}^{\mathrm{T}}}\frac{\partial\mathbf{Q}^{(k)}}{\partial\mathbf{c}^{(k)T}} = g'(\delta\mathbf{x}^{(k+1)} \odot \sigma'(\mathbf{a}^{(k+1)}))\alpha^{(k)}\delta\mathbf{x}^{(k)\mathrm{T}} = \alpha^{(k)}g'(\delta\mathbf{a}^{(k+1)})\delta\mathbf{x}^{(k)\mathrm{T}}. \tag{17}$$

When ignoring the magnitude of the gradients we have $\frac{\partial\mathbf{Q}}{\partial\mathbf{W}} \approx \frac{\partial\mathbf{Q}^{(k)}}{\partial\mathbf{W}} \approx \frac{\partial\mathbf{Q}^{(k+1)}}{\partial\mathbf{W}}$. If $\mathbf{x}^{(i)}$ is projected onto $\delta\mathbf{x}^{(i)}$ we have $\mathbf{x}^{(i)} = \frac{\mathbf{x}^{(i)\mathrm{T}}\delta\mathbf{x}^{(i)}}{\|\delta\mathbf{x}^{(i)}\|^2}\delta\mathbf{x}^{(i)} + \mathbf{x}^{(i)}_{res}$. The prescribed update for $\mathbf{W}$ is:

$$\delta\mathbf{W} = -\delta\mathbf{x}^{(k+1)}\frac{\partial\mathbf{x}^{(k+1)}}{\partial\mathbf{W}} = -g'(\delta\mathbf{x}^{(k+1)} \odot \sigma'(\mathbf{a}^{(k+1)}))\mathbf{x}^{(k)\mathrm{T}}$$

$$= -g'(\delta\mathbf{a}^{(k+1)})\mathbf{x}^{(k)\mathrm{T}} = -g'(\delta\mathbf{a}^{(k+1)})(\alpha^{(k)}\mathbf{P}^{(k)}\delta\mathbf{x}^{(k)} + \mathbf{x}^{(k)}_{res})^{\mathrm{T}} \tag{18}$$

$$= -\alpha^{(k)}\mathbf{P}^{(k)}g'(\delta\mathbf{a}^{(k+1)})\delta\mathbf{x}^{(k)\mathrm{T}} - g'(\delta\mathbf{a}^{(k+1)})\mathbf{x}^{(k)\mathrm{T}}_{res} = -\mathbf{P}^{(k)}\frac{\partial\mathbf{Q}^{(k)}}{\partial\mathbf{W}} - g'(\delta\mathbf{a}^{(k+1)})\mathbf{x}^{(k)\mathrm{T}}_{res}$$

It is obvious that $\mathbf{Q}^{(k)}$ and $\mathbf{Q}^{(k+1)}$ can be maximized when the component of $\frac{\partial\mathbf{Q}^{(k)}}{\partial\mathbf{W}}$ in $\delta\mathbf{W}$ is maximized by minimizing $\mathbf{P}^{(k)}$. The gradient to minimize $\mathbf{P}^{(k)}$ is the prescribed update $-\delta\mathbf{x}^{(k)}$. The

angle between $\delta\mathbf{x}^{(k)}$ and the gradient of BP $\mathbf{c}^{(k)}$ is within $90°$ if $\mathbf{Q}^{(k)} > 0$ because the cosine of the two vector is $\frac{\mathbf{Q}^{(k)}}{\|\mathbf{c}^{(k)}\|} > 0$, and it also indicates that $\mathbf{c}^{(k)}$ is nonzero and therefore in a descending trend. Consequently, $\delta\mathbf{x}^{(k)}$ will be oriented towards a descending direction since any vector that lies within $90°$ of the steepest descent direction will similarly point downwards, in other words, for GCN, a broad spectrum of asymmetric feedback paths can offer a descending gradient direction for a hidden layer as long as $\mathbf{Q}^{(i)} > 0$.

From Eq. 5, it is obvious one advantage of our method is that $\delta\mathbf{x}^{(i)}$ is non-zero for any non-zero error $\mathbf{e}$, as a randomly generated matrix $\mathbf{B}^{(i)}$ is highly likely to be of full rank. Ensuring $\delta\mathbf{x}^{(i)}$ is non-zero is crucial for achieving $\mathbf{Q}^{(i)} > 0$. Maintaining static feedback across training helps preserve the characteristic, and also simplifies the process of maximizing $\mathbf{Q}^{(i)}$ due to the more consistent direction to $\delta\mathbf{x}^{(i)}$. Our method shows better biological plausibility in GNN training, which introduces asymmetric feedback paths to take place of BP, not only solving the weight transport problem and partially solving the update locking problem, but also releasing the requirement to store neural activations and accumulated gradients for backward propagation.

## A.4 Datasets Statistics

We evaluate our method on 10 benchmark datasets across domains: citation networks (Cora, Cite-Seer, PubMed) [Sen et al., 2008, Yang et al., 2016], Amazon co-purchase graph (Photo, Computer) [McAuley et al., 2015], Wikipedia graphs (Chameleon, Squirrel) [Rozemberczki et al., 2021], actor co-occurrence graph (Actor) [Pei et al., 2019] and webpage graphs from WebKB (Texas, Cornell) [Pei et al., 2019]. The datasets adopted are representative which describe diverse real-life scenarios. Some of them are highly homophilic while others are heterophilic. Note that for Chameleon and Squirrel, we use the filtered version from Platonov et al. [2022], as the original version from [Rozemberczki et al., 2021] may contain duplicated nodes. The detailed statistics of the datasets are summarized in Tab. 5. We compute the node homophily for each dataset using the method proposed by Pei et al. [2019], referred to as the homophily value.

Table 5: Datasets statistics.

|           | Cora  | CiteSeer | PubMed | Computer | Photo  | Chameleon | Squirrel | Actor | Texas | Cornell |
|-----------|-------|----------|--------|----------|--------|-----------|----------|-------|-------|---------|
| Nodes     | 2708  | 3327     | 19717  | 13752    | 7650   | 890       | 2223     | 7600  | 183   | 183     |
| Edges     | 5278  | 4552     | 44324  | 245861   | 119081 | 17708     | 93996    | 26659 | 279   | 277     |
| Features  | 1433  | 3703     | 500    | 767      | 745    | 2325      | 2089     | 932   | 1703  | 1703    |
| Classes   | 7     | 6        | 5      | 10       | 8      | 5         | 5        | 5     | 5     | 5       |
| Homophily | 0.825 | 0.706    | 0.792  | 0.785    | 0.836  | 0.244     | 0.190    | 0.220 | 0.057 | 0.301   |

- **Cora**, **CiteSeer** and **PubMed** [Sen et al., 2008] are three classic homophilic citation networks. In these networks, nodes correspond to academic papers, and edges signify the citation links between papers. The node features are derived from bag-of-word representations of the papers, and the labels categorize each paper into specific research topics.

- **Computer** and **Photo** [McAuley et al., 2015] are segments of the Amazon co-purchase graph, where nodes represent goods, edges indicate that two goods are frequently bought together, node features are bag-of-words encoded product reviews, and class labels are given by the product category.

- **Chameleon** and **Squirrel** [Rozemberczki et al., 2021] are two heterophilic networks derived from Wikipedia. In these networks, nodes represent Wikipedia web pages, and edges correspond to hyperlinks between these pages. The features are comprised of informative nouns extracted from the Wikipedia content, while the labels reflect the average traffic of each web page.

- **Actor** [Pei et al., 2019] is a heterophilic actor co-occurrence network where nodes represent actors, and edges signify that two actors have appeared together in the same movie. The features are derived from keywords found on the actors' Wikipedia pages, while the labels consist of significant words associated with each actor.

- **Cornell** and **Texas** [Pei et al., 2019] are heterophilic networks from the WebKB1 project representing computer science departments at three universities. Nodes are departmental web pages, edges represent hyperlinks, features are derived using bag-of-words, and labels categorize page types. These networks illustrate heterophilic connections where linked pages often differ in type.

## A.5 Experimental Settings

We conduct the semi-supervised node classification task using a basic GCN model, where the node set is randomly divided into the train/validation/test set with 60%/20%/20%. For fairness, we generate 10 random splits using different seeds and evaluate all approaches on these identical splits, reporting the average performance for each method. Our method is compared with five baseline training strategies including the traditional backpropagation (BP) [Rumelhart et al., 1986], PEPITA [Dellaferrera and Kreiman, 2022], two versions of the forward-forward algorithm (*abbr.* FF+LA, FF+VN) [Hinton, 2022, Park et al., 2023], two versions of the cascaded forward algorithm (*abbr.* CaFo+MSE, CaFo+CE) [Zhao et al., 2023] and the FORWARDGNN Single-Forward algorithm (*abbr.* SF) [Park et al., 2023] specifically designed for GNNs. To ensure fairness, we train all approaches using the same GCN architecture, which includes 3 graph convolutional layers and 64 hidden units—sufficiently representative for all datasets. We employ Adam as the optimization algorithm, refraining from using any regularization techniques other than an appropriate L2 penalty specific to Adam. The evaluation metric used is accuracy (acc), presented with a 95% confidence interval.

Table 6: Hyper-parameters of proposed method on real-world datasets.

|  | Learning rate | Hidden unit | $\alpha$ | Iteration epoch | $\epsilon$ | Weight decay | Training epoch |
|---|---|---|---|---|---|---|---|
| Cora | 0.01 | 64 | 0.1 | 50 | 0.5 | 0.0005 | 1000 |
| CiteSeer | 0.01 | 64 | 0.01 | 200 | 0.5 | 0.0005 | 1000 |
| PubMeb | 0.01 | 64 | 0.1 | 200 | 0.5 | 0.0005 | 1000 |
| Photo | 0.001 | 64 | 0.01 | 50 | 0.5 | 0.0005 | 1000 |
| Computer | 0.001 | 64 | 0.01 | 50 | 0.5 | 0.0005 | 1000 |
| Texas | 0.01 | 64 | 0.9 | 50 | 0.5 | 0.0 | 1000 |
| Cornell | 0.01 | 64 | 0.5 | 50 | 0.5 | 0.0 | 1000 |
| Actor | 0.01 | 64 | 0.5 | 50 | 0.5 | 0.0 | 1000 |
| Chameleon | 0.01 | 64 | 0.5 | 200 | 0.5 | 0.0 | 1000 |
| Squirrel | 0.01 | 64 | 0.5 | 200 | 0.5 | 0.0 | 1000 |

The codes of DFA-GNN are based on the GNNs in the PyTorch version by Deep Graph Library (DGL) [Wang et al., 2019]. To generate pseudo errors, we search the optimal $\alpha$ in Eq. 8 within {0.001, 0.01, 0.1, 0.3, 0.5, 0.7, 0.9}, the iteration epoch for Eq. 8 within {5, 10, 30, 50, 100, 150, 200}, $\epsilon$ in Eq. 9 within {0.3, 0.5, 0.7, 0.9}. For the training of our method, we search the learning rate within {0.001, 0.01, 0.1} and weight decay within {0.0005, 0}. All the experiments are run on AMD EPYC 7542 32-Core Processor with Nvidia GeForce RTX 3090. We list the hyper-parameter values used in our model in Tab. 6.

## A.6 Time Comparison

Table 7: Average running time per epoch (s). For layer-wise training methods like PEPITA, CaFo, FF and SF, the total time taken by each layer per epoch is reported.

| | BP | PEPETA | CaFo+CE | FF+LA | FF+VN | SF | ours |
|---|---|---|---|---|---|---|---|
| Cora | $7.56e^{-3}$ | $8.73e^{-3}$ | $7.61e^{-1}$ | $3.14e^{-1}$ | $2.83e^{-1}$ | $5.49e^{-2}$ | $5.66e^{-2}$ |
| CiteSeer | $1.06e^{-2}$ | $1.11e^{-2}$ | $7.68e^{-1}$ | $2.59e^{-1}$ | $2.61e^{-1}$ | $6.88e^{-2}$ | $5.68e^{-2}$ |
| PubMed | $1.07e^{-2}$ | $1.07e^{-2}$ | $8.24e^{-1}$ | $6.94e^{-1}$ | $7.61e^{-1}$ | $5.34e^{-1}$ | $6.76e^{-2}$ |
| Photo | $8.74e^{-3}$ | $1.03e^{-2}$ | $7.98e^{-1}$ | $2.11$ | $1.91$ | $4.87e^{-1}$ | $5.81e^{-2}$ |
| Computer | $1.08e^{-2}$ | $1.05e^{-2}$ | $7.80e^{-1}$ | $4.82$ | $4.14$ | $7.61e^{-1}$ | $6.29e^{-2}$ |
| Texas | $6.13e^{-3}$ | $1.07e^{-2}$ | $8.05e^{-1}$ | $1.47e^{-1}$ | $1.56e^{-1}$ | $6.88e^{-2}$ | $5.60e^{-2}$ |
| Cornell | $5.42e^{-3}$ | $1.06e^{-2}$ | $7.46e^{-1}$ | $1.51e^{-1}$ | $1.24e^{-1}$ | $3.59e^{-2}$ | $5.53e^{-2}$ |
| Actor | $9.45e^{-3}$ | $1.03e^{-2}$ | $7.83e^{-1}$ | $6.84e^{-1}$ | $6.71e^{-1}$ | $2.80e^{-1}$ | $5.80e^{-2}$ |
| Chameleon | $6.24e^{-3}$ | $1.13e^{-2}$ | $7.97e^{-1}$ | $2.28e^{-1}$ | $2.09e^{-1}$ | $6.88e^{-2}$ | $5.61e^{-2}$ |
| Squirrel | $7.77e^{-3}$ | $1.20e^{-2}$ | $7.78e^{-1}$ | $5.82e^{-1}$ | $5.05e^{-1}$ | $1.21e^{-1}$ | $5.79e^{-2}$ |

## A.7 Comparison of BP and DFA with Pseudo-Error Generation

Although the pseudo-error generation process is essential for our method, it is also an optional choice for backpropagation. We integrate this component into BP, and the experimental results from Tab. 8 show that although this component contributes to DFA in our method, it does not positively enhance BP overall. Even with pseudo-error generation, BP cannot outperform our method. This observation indicates that direct feedback of errors may benefit more from pseudo-errors rather than the layer-by-layer backward pass.

Table 8: Results of BP and DFA with pseudo-error generation spreading: mean accuracy (%)±95% confidence interval.

| | Cora | CiteSeer | PubMed | Photo | Computer | Actor | Chameleon | Squirrel |
|---|---|---|---|---|---|---|---|---|
| BP | 86.04 | 78.20 | 85.24 | 93.03 | **89.48** | 31.94 | 41.28 | 37.81 |
| | ±0.62 | ±0.57 | ±0.28 | ±0.59 | ±0.37 | ±0.88 | ±2.29 | ±0.71 |
| BP+EG | 87.41 | 80.24 | 84.74 | 91.95 | 87.78 | 31.53 | 38.71 | 35.78 |
| | ±0.80 | ±1.11 | ±0.41 | ±0.41 | ±0.51 | ±1.91 | ±2.23 | ±1.52 |
| | (1.37↑) | (2.04↑) | (0.50↓) | (1.08↓) | (1.70↓) | (0.41↓) | (2.57↓) | (2.03↓) |
| DFA | 83.02 | 78.17 | 82.92 | 91.75 | 84.02 | 30.72 | 39.09 | 34.61 |
| | ±1.36 | ±0.71 | ±0.42 | ±0.48 | ±1.54 | ±1.72 | ±1.17 | ±1.01 |
| DFA+EG (ours) | **87.72** | **80.49** | **86.28** | **93.04** | 86.72 | **34.07** | **41.19** | **38.17** |
| | ±1.63 | ±0.41 | ±0.67 | ±0.31 | ±0.68 | ±0.75 | ±1.56 | ±2.21 |
| | (4.70↑) | (2.32↑) | (3.36↑) | (1.29↑) | (2.70↑) | (3.35↑) | (2.10↑) | (3.56↑) |

## A.8 DFA-GNN with Alternative Activation Functions

We conduct experiments for our method with four different activation functions (*i.e.*, Sigmoid, Tanh, ELU and LeakyReLU) as shown in Tab. 9. The results demonstrate our method is well integrated with different activation functions and derives consistently good results.

Table 9: Results of our method with different activation functions.

| | ours+Sigmoid | ours+Tanh | ours+ReLU | ours+ELU | ours+LeakyReLU (slope=0.2) |
|---|---|---|---|---|---|
| Cora | 87.68±1.75 | 88.17±1.66 | 87.72±1.63 | 87.93±1.75 | 87.64±1.65 |
| CiteSeer | 79.94±0.74 | 80.01±0.75 | 80.49±0.41 | 81.11±0.86 | 80.43±0.75 |
| PubMed | 84.57±0.32 | 85.57±0.40 | 86.28±0.67 | 85.53±0.48 | 84.99±0.39 |

