# OpenReview forum: "DFA-GNN: Forward Learning of Graph Neural Networks by Direct Feedback Alignment"
_NeurIPS.cc/2024/Conference — NeurIPS 2024 poster_

### Official Review · Reviewer_mUbj · 2024-07-12

**Soundness:** 2
**Presentation:** 3
**Contribution:** 2
**Rating:** 4
**Confidence:** 4

**Summary:**

The paper introduces DFA-GNN, a new framework designed to train Graph Neural Networks (GNNs) using Direct Feedback Alignment (DFA). Traditional methods like backpropagation (BP), though effective, have several limitations, including inefficiency, scalability issues, and a lack of biological plausibility. DFA-GNN aims to address these problems by utilizing a forward training mechanism specifically tailored for the unique challenges posed by graph data.

**Strengths:**

The paper is well written and easy to follow.

Performance: DFA-GNN achieves better performance compared to other non-BP methods and even standard BP methods.

Robustness: The framework exhibits resilience to noise and different types of attacks, ensuring its reliability in various scenarios.

Efficiency: By enabling parallel gradient computation, DFA-GNN improves the efficiency of the training process.

**Weaknesses:**

The novelty of this paper is somewhat limited as it primarily extends the principles of DFA to GNNs. The core ideas are based on the foundations laid out in the Shalizi et al. (2013) paper.

The paper lacks a discussion on the computational complexity of the proposed algorithm.

There is a notable variation in the performance improvements observed across different datasets. For instance, DFA-GNN shows a significant accuracy improvement on the Texas and Cornell datasets, while the improvements on other datasets are relatively modest. The paper does not adequately explain the reasons behind this variation in performance.

The dataset used for experiments is quite small.

**Questions:**

Why the proposed method doesnot outperform on the computer and chameleon dataset.

How does DFA-GNN scale when applied to larger datasets e.g. flicker, Reddit, OBGN ?

What is the time complexity of the proposed method?

**Limitations:**

No societal impacts.

---

> ### Author Rebuttal · Authors · 2024-08-04
>
> We are grateful to the reviewer for the time taken to assess our work and for the valuable feedback. We address each point individually. “W/Q” numbers the weakness or question followed by our response.
>
> $\textbf{\large Response to W1:} $
>
> Thanks for your comments. Our method builds on some ingenious previous studies, especially $\textit{Nøkland A. Direct feedback alignment provides learning in deep neural networks, NeurIPS 2016}$. However, our work is the first to systematically study DFA's applicability to GNNs with extensive ablation studies and to provide update formulas and convergence proof for DFA in GNNs. The addition of pseudo-error generation is another highlight that is entirely novel for DFA. Our work demonstrates the exciting potential of non-BP training in graph deep learning, offering potential directions for addressing challenging issues such as oversquashing, topology distortion, and label scarcity, which warrant further exploration in the future.
>
> $\textbf{\large Response to W2 and Q3:} $
>
> The time complexity of our method depends on the architecture of the GNN models. Given specific GNN layers with a defined forward propagation formula, the computational time cost of our method includes four parts: forward propagation (Eq.1), pseudo-error generating (Eqs.7, 8), node filtering (Eq.8) and direct feedback alignment (Eq.10). The complexity of forward propagation is the same as BP because our method makes no change in this process. Supposing the dimension of node features and hidden units is $d$, for the GCN model formulated in Eq.1 whose forward propagation formula for $i$-th layer is written as $\textbf{X}^{(i)}=relu(\textbf{S}\textbf{X}^{(i-1)}\textbf{W}^{(i-1)})$, the complexity is $\mathcal{O}(n^{2}d+nd^{2})$, where $n$ denotes the number of nodes. Since $n\gg d$ holds in most datasets, the complexity can be approximated as $\mathcal{O}(n^{2}d)$. For direct feedback alignment, according to Eq.10, the time complexity for each layer is $\mathcal{O}(n_{f}^{2}c+dn_{f}c+d^{2}c)$ and is further approximated as $\mathcal{O}(n_{f}^{2}c)$ due to $n_{f}\gg d \gg c$, where $n_{f}$ is the number of nodes after filtering and $n_{f}<n$, $c$ the number of category. In comparison, BP for GCN model has a complexity of $\mathcal{O}(n^{2}d+nd^{2}+d^{3})$ for each layer, which could be approximated as $\mathcal{O}(n^{2}d)$. Although both the methods have a quadratic complexity related to the number of nodes, our method makes parallel update for each layer while BP has to run update layer by layer.
>
> The other two components pseudo-error generator and node filter bring additional time cost to our method compared with BP. According to Eqs.7 and 8, the time complexity of the two parts is $\mathcal{O}(tcn^{2})$ and $\mathcal{O}(cn)$, respectively, where $t$ denotes the iteration epochs for error spreading. Although the two introduced parts result in increased time cost, they are essential for generalizing DFA to GNNs. The ablation study in Tab.2 demonstrates the significant contribution of these two components. Overall, our methods has the same quadratic time complexity as BP on GCN, which explains why our method has a consistent five to ten times consumption per epoch compared with BP, rather than an exponential difference, regardless of the dataset size.
>
> $\textbf{\large Response to W3 and Q1:} $
>
> Thanks for the suggestion, and we will add related discussion in the revised version.
>
> Our method shows a much more significant accuracy improvement on relatively small datasets, such as Texas and Cornell. The reason mainly lies in that GNNs trained by BP often suffer from heavy overfitting due to the scarcity of supervision on small datasets, especially for semi-supervised tasks. By contrast, the pseudo error generated in our method not only is for gradient computation when adapting DFA to GNNs, but also serves as an data augmentation strategy, enhancing the robustness of GNNs (shown in Fig.3) and improving the ability for handling overfitting with scarce supervision. The gains derived from the pseudo-error strategy is not so significant on larger datasets (e.g., PubMed, Photo), but it still brings overall positive effects to our methods.
>
> For Computer and Chameleon, our method does not outperform BP, however, the accuracy gap on Chameleon is quite small (only 0.09\%). The testing accuracy depends on both the training algorithm and GNN model. Although our method underperforms BP for GCN on Computer, however, our method outperforms BP for SGC, GraphSage, APPNP and ChebyNet on this dataset as shown in Tab.3. Generally, our non-BP method has comparable results compared with BP and even show better performance on most experimental trails, which indicates our non-BP training method has the inspiring potential to become a substitute for BP in the field of graph deep learning.
>
> $\textbf{\large Response to W4 and Q2:} $
>
> Our method is well-suited for large datasets. When the graph scale is large, for memory saving we can use edge indices instead of an adjacency matrix to store the graph. For forward propagation (Eq.1), the space requirement for neighbor aggregation can be reduced from $\mathcal{O}(n^{2}+nd)$ to $\mathcal{O}(|\mathcal{E}|+nd)$, where $|\mathcal{E}|$ denotes the number of edges. For direct feedback alignment (Eq.10), as $(\textbf{S}^{\text{T}})^{k}\hat{\textbf{E}}$ is exactly the aggregation of errors for $k$ times, the space requirement can be reduced to $\mathcal{O}(|\mathcal{E}|+nc)$, without the need to store the $k$-th power of the adjacency matrix. Similarly, space requirement reduction can also be achieved in the node filtering process.
>
> As for prediction accuracy, the experimental results on Flickr, Reddit, and ogbn-arxiv (shown in **Tab.X** in the supplementary PDF) demonstrate that our method obtains comparable results to BP (within 1% accuracy) and outperforms other non-BP methods with only a small memory cost (2049 MiB, 11675 MiB, 2277 MiB for the three datasets, respectively).

---

> > ### Comment · Reviewer_mUbj · 2024-08-13
> > **Response to Author's Rebuttal**
> >
> > I would like to thank the authors for your response.  However, I still have concerns in these points:
> > 1.I could not find a table comparing the time efficiency of your method with state of the art method.
> > 2. I have concerns about the time complexity. The additional step also contribute to an O(n²) complexity? How much it isfaster than standard BP methiod?
> > 3. The novelty of your approach seems limited, as it primarily involves integrating DFA into GNNs.

---

> ### Author Response · Authors · 2024-08-13
>
> Thank you once again for your valuable feedback on our paper.
>
> 1. The time comparison of our method is presented in Table 6 of Appendix A.6 in the original manuscript (for your convenience, we have displayed the table below). Here, we compare the training time per epoch of our method with BP and other state-of-the-art non-BP methods. For the semi-supervised node classification task, all methods are trained for 1000 epochs. During training, we calculate the prediction accuracy on the validation nodes after each epoch and save the model that achieves the highest validation accuracy. This saved model is then used to make predictions on the test nodes. This approach is standard in semi-supervised graph learning. Therefore, the overall training cost is simply one thousand times the numbers shown in Table 6.
>
>
> 2. As noted in our response to W2 and Q3, both BP and the DFA in our method have a complexity of $\mathcal{O}(n^{2})$. Besides, our method introduces an additional complexity of $\mathcal{O}(n^{2})$ due to the pseudo-error generator. As a result, the overall complexity for both our method and BP remains the same $\mathcal{O}(n^{2})$. This explains why our method consistently takes five to ten times longer per epoch compared with BP as shown in Table 6, rather than showing an exponential difference, regardless of dataset size. While BP benefits from decades of research and strong software and hardware support, our method has not yet reached comparable time efficiency of BP. However, our method demonstrates superior time efficiency compared with most non-BP state-of-the-art methods, and the current time cost difference compared with BP is not substantial. Additionally, the parallel update strategy employed by our method for each layer offers considerable potential for parallel computing, which could be further explored to enhance time efficiency.
>
> 3. We theoretically derive formulas to integrate the direct feedback alignment mechanism into GNNs, as DFA for fully connected layers is not directly applicable to graph data. Additionally, we provide theoretical proof of the convergence of our proposed method (as shown in Section 4.3 and Appendix A.3). Our work highlights the promising potential of non-BP training in graph deep learning and opens up avenues for tackling some challenges within graphs, which merit further exploration in future research.
>
>
> If our rebuttal has satisfactorily addressed your concerns, we would greatly appreciate it if you could consider reevaluating the score of our paper. Regardless of your decision, we are truly grateful for your guidance and the time you have invested in reviewing our work.
>
> Thank you again for your attention and support.
>
> Best regards,
>
> Authors
>
>
> $\newline$
>
> **Table 6: Average running time per epoch (s). For layer-wise training methods like PEPITA, CaFo, FF, and SF, the total time taken by each layer per epoch is reported.**
>
> | Datasets  | BP      | PEPITA  | CaFo+CE | FF+LA  | FF+VN  | SF      | ours    |
> |----------|---------|---------|---------|--------|--------|---------|---------|
> | Cora     | 7.56e-3 | 8.73e-3 | 7.61e-1 | 3.14e-1 | 2.83e-1 | 5.49e-2 | 5.66e-2 |
> | CiteSeer | 1.06e-2 | 1.11e-2 | 7.68e-1 | 2.59e-1 | 2.61e-1 | 6.88e-2 | 5.68e-2 |
> | PubMed   | 1.07e-2 | 1.07e-2 | 8.24e-1 | 6.94e-1 | 7.61e-1 | 5.34e-1 | 6.76e-2 |
> | Photo    | 8.74e-3 | 1.03e-2 | 7.98e-1 | 2.11   | 1.91   | 4.87e-1 | 5.81e-2 |
> | Computer | 1.08e-2 | 1.05e-2 | 7.80e-1 | 4.82 | 4.14   | 7.61e-1 | 6.29e-2 |
> | Texas    | 6.13e-3 | 1.07e-2 | 8.05e-1 | 1.47e-1 | 1.56e-1 | 6.88e-2 | 5.60e-2 |
> | Cornell  | 5.42e-3 | 1.06e-2 | 7.46e-1 | 1.51e-1 | 1.24e-1 | 3.59e-2 | 5.53e-2 |
> | Actor    | 9.45e-3 | 1.03e-2 | 7.83e-1 | 6.84e-1 | 6.71e-1 | 2.80e-1 | 5.80e-2 |
> | Chameleon| 6.24e-3 | 1.13e-2 | 7.97e-1 | 2.28e-1 | 2.09e-1 | 6.88e-2 | 5.61e-2 |
> | Squirrel | 7.77e-3 | 1.20e-2 | 7.78e-1 | 5.82e-1 | 5.05e-1 | 1.21e-1 | 5.79e-2 |

---

> > ### Comment · Reviewer_mUbj · 2024-08-13
> > **Response to Author's Rebuttal**
> >
> > According to this statement  although both the method have quadratic complexity, "our method make parallel update for each layer while BP has to update layer by layer". Therefore, the proposed method should theoretically be faster than BP. However, this is not evident from the results in Table 6 of the appendix. Could you please clarify this discrepancy?

---

> > > ### Author Response · Authors · 2024-08-14
> > >
> > > Dear Reviewer,
> > >
> > > Thank you for your valuable feedback on our paper. We have carefully addressed your comments in our response and submitted it for your review. As the deadline for responses is approaching, we wanted to kindly inquire if there are any additional questions or points of clarification you would like us to address. If any questions are not answered or our response is unclear, we would appreciate the opportunity to communicate further with you.
> > >
> > > We sincerely appreciate your time and effort in reviewing this manuscript.
> > >
> > > Best regards,
> > >
> > > Authors

---

> ### Author Response · Authors · 2024-08-13
>
> We apologize for the confusing expression. The standalone direct feedback alignment (DFA, $\mathcal{O}(n^{2})$) as formulated in Eqs. 5 and 6 of the manuscript can potentially be faster than backpropagation (BP, $\mathcal{O}(n^{2})$) since DFA directly calculates the parameter updates for each layer from the loss, allowing for improved training efficiency through parallelization. However, to generalize DFA for graph data, we introduced the pseudo-error generator as detailed in Sec. 4.2. Consequently, the main time cost of our method arises not only from DFA but also from the pseudo-error generation (Eqs. 7, 8) and node filtering (Eq. 9).
>
> According to Eqs. 7 and 8, the time complexity of these two components is $\mathcal{O}(tcn^{2})$ and $\mathcal{O}(cn)$, respectively, where $t$, $c$, and $n$ denote the iteration epochs for error spreading, the number of categories, and the number of nodes, respectively (In our earlier analysis, we approximated the complexity of these components as $\mathcal{O}(n^{2})$ and $\mathcal{O}(n)$ to highlight that our method maintains the same quadratic complexity with respect to $n$, the most significant factor). When these two components are added, our method consistently incurs a time cost that is 5 to 10 times larger than that of BP even though they have the same order of complexity. Although these additional components increase the time cost, they are crucial for adapting DFA to GNNs and handling graph-structured data. The ablation study in Tab. 2 highlights the substantial contribution of these components.
>
> Although our method has achieved significant practical time efficiency advantages over existing non-BP methods, we will continue to explore ways to further improve the time efficiency of our training algorithm in future work to bring it closer to that of BP, and we will revise the related sections of the article to reflect these nuances more accurately.
>
>
> Thank you again for your attention and support.
>
> Best regards,
>
> Authors

---

### Official Review · Reviewer_buRu · 2024-07-13

**Soundness:** 3
**Presentation:** 3
**Contribution:** 2
**Rating:** 6
**Confidence:** 4

**Summary:**

The authors propose to apply the Direct Feedback Alignment (DFA) algorithm for backpropagation-free training to graph neural networks. The DFA algorithm is combined with a pseudo-error generation mechanism to provide additional error signals for missing targets in the setting of semi-supervised node classification. The experimental results with a 3-layer GCN model over 10 commonly adopted benchmarks show the proposed model performing generally better than backpropagation and other training algorithms. The ablation analysis confirms that improvements by the proposed DFA-GNN approach can be ascribed to the pseudo-error generation mechanism.

**Strengths:**

**Originality**
The application of DFA to GNNs was already introduced in *J. Launay, I. Poli, F. Boniface, and F. Krzakala, “Direct feedback alignment scales to modern deep learning tasks and architectures,” NeurIPS 2020*. However, the addition of pseudo-error generation and the extensive experiments are entirely novel contributions.

**Quality**
The submission is technically sound and its claims are well supported.

**Clarity**
The paper is clearly written and well organized.

**Significance**
The results of this paper can contribute to the research on addressing the training challenges of deep graph neural networks.

**Weaknesses:**

1. The computational cost per epoch of the proposed algorithm is 5 to 10 times larger than backpropagation.
2. No comparison of the overall training costs, including the different number of epochs required for training convergence among the different methods.
3. The ablation indicates that the accuracy improvements are ascribed to the pseudo-error propagation mechanism. As the latter could be applied also to e.g. backpropagation, such experimental evaluation would have provided further interesting insights.
4. In the experiments the model is limited to 3 layers; deeper models would have provided more interesting insights, as depth increases the issues connected with gradient backpropagation. (Long-range graph benchmarks such as those from *V. P. Dwivedi et al., “Long Range Graph Benchmark,” in NeurIPS 2022 Track on Datasets and Benchmarks* could be adopted in this case.)
5. Only node classification tasks are considered, no graph-level or edge-level tasks.

**Questions:**

1. Why is pseudo-error generation spreading, which must be done at teach gradient update iteration, instead of augmenting training labels via label spreading (plus the masking of eq. (9)), which would be only required once?
2. How would the other training methods such as backpropagation perform with pseudo-error generation spreading?
3. How many epochs are required for training convergence by DFA-GNN? How do they compare with the number of epochs required by the other training algorithms?

**Limitations:**

The limitations of this submission have not been properly discussed. For example, in lines 278-280, it is stated that the "method has a higher training time consumption compared with BP, primarily due to the additional time needed for generating pseudo errors and filtering masks". However, in the conclusions, it is claimed that *efficiency* is one of the advantages of the proposed method.

---

> ### Author Rebuttal · Authors · 2024-08-04
>
> We appreciate your thoughtful comments and positive assessment of our work. After carefully reviewing your feedback, below we provide answers to the comments you raised.
>
> $\textbf{\large Response to W1:}  $
>
> The standalone direct feedback alignment formulated in Eqs.5 and 6 could be faster than BP. However, to generalize DFA to graph data we introduce the pseudo-error generator as elaborated in Sec.4.2, and the main time cost comes from the pseudo-error generation (Eqs.7, 8) and node filtering (Eq.9). According to Eqs.7 and 8, the time complexity of the two parts is $\mathcal{O}(tcn^{2})$ and $\mathcal{O}(cn)$, respectively, where $t, c, n$ denote the iteration epochs for error spreading, the number of categories and the number of nodes, respectively. Although the two introduced parts result in increased time cost, they are essential for generalizing DFA to GNNs. The ablation study in Tab.2 demonstrates the significant contribution of these two components.
>
> $\textbf{\large Response to W2 and Q3:}  $
>
> Thanks for your comments. For our semi-supervised node classification task, we train all the methods for 1000 epochs, calculating the prediction accuracy for validation nodes after each epoch and saving the model that achieves the highest validation accuracy. We use the saved model to make predictions for the test nodes. This is a common experimental setting in semi-supervised graph learning. Thus, the overall training cost is simply one thousand times the numbers shown in Tab.6.
>
> To better illustrate the training convergence of our DFA-GNN, we plot the training and validation accuracy of the proposed DFA-GNN and BP over training epochs on three datasets, as shown in **Fig. 4** in the supplementary PDF. In general, our method shows similar convergence as BP. For both BP and our method, the convergence of validation accuracy occurs much earlier than that of training accuracy due to overfitting. The validation convergent epoch of DFA-GNN is nearly the same as BP on Cora and CiteSeer (around 100 epochs), while it is 100 epochs later than BP on PubMed (around 200 epochs). Our method achieves better validation accuracy on all these datasets and suffers less from overfitting compared with BP.
>
> In terms of training accuracy, our method exhibits slightly lower convergent value and lower convergence speed than BP.  The reason is that our method considers both the errors of labeled nodes and pseudo errors of unlabeled nodes as supervision signals.
> However, this does not prevent our method from achieving better validation results.
>
> We do not include other non-BP methods in Fig. 4 because these methods (i.e., SF, FF, CaFo, etc.) follow a layer-by-layer update according to local loss. Therefore, the convergence behaviors varies across different layers, and it is hard to directly visualize them in a global view.
>
> $\textbf{\large Response to W3 and Q2:}  $
>
> Thanks for your interesting suggestion. Although the pseudo-error generation process is essential for our method, it is also an optional choice for BP. We try integrating this component into BP, and the experimental results in **Tab.IX** in the supplementary PDF show that this component does not positively enhance BP in general. It only contributes to BP on Cora and CiteSeer while degrading the performance on the other six datasets. By contrast, it benefits DFA remarkably across all datasets and makes our method finally outperform BP in most scenarios. The different suitabilities of the pseudo-error generation to BP and DFA mainly lie in that DFA is more noise-tolerant than BP where the generated pseudo error always contains different degrees of noise, which has been discussed in some previous studies [a,b].
>
> [a]. R. Ohana et. al., Photonic Differential Privacy with Direct Feedback Alignment. NeurIPS 2021.
>
> [b]. J. Lee and D. Kifer. Differentially Private Deep Learning with Direct Feedback Alignment. arXiv 2020.
>
> $\textbf{\large Response to W4:}  $
>
> Thanks for the suggestion. We plot the prediction accuracy with the number of layers increasing from 2 to 10, as shown in **Fig. 5** in the supplementary PDF. Both the BP and our method show a decrease in accuracy, but our method is less negatively affected and consistently outperforms BP. This superiority is derived from the fact that our method adopts direct feedback of errors for the update of each layer, reducing the problem of oversmoothing and other gradient issues caused by backward gradient accumulation. This observation indicates the potential of our method for deep GNN training.
>
> $\textbf{\large Response to W5:}  $
>
> Thanks for your suggestion, we will explore the potential of DFA-GNN for graph-level and edge-level tasks in future work.
>
> $\textbf{\large Response to Q1:}  $
>
> The pseudo error is essential for our method because the parameter update formula (Eqs.5 and 6) for each layer requires the error of all nodes. Spreading labels may not be well-suited to our update mechanism. Additionally, since the pseudo error depends on the prediction of each epoch, updating pseudo errors in real-time based on the results of forward propagation in each epoch is a more reasonable choice. Augmenting only once in an early epoch may lead to performance degradation of model.
>
> $\textbf{\large Response to Limitations:} $
>
> We apologize for the confusing expression. The efficiency advantage of our method is mainly reflected in the comparison with other non-BP algorithms, showing significant time superiority over CaFo, FF, and SF, as presented in Tab.6. As BP has been extensively researched for decades and benefits from robust software and hardware support, our method has not yet achieved comparable time efficiency of BP. However, the parallel update strategy for each layer in our method makes it more conducive to parallel computing, which has the potential to further improve time efficiency.  We will revise the conclusion section of the article to make it more rigorous.

---

> > ### Comment · Reviewer_buRu · 2024-08-12
> >
> > Thanks to the authors for their response. I leave my score unchanged.

---

> > > ### Author Response · Authors · 2024-08-13
> > >
> > > Thank you for your valuable feedback and for acknowledging our efforts. We respect your decision and sincerely appreciate your constructive critique. Your insights have been instrumental in helping us refine our work, and we are grateful for your thorough and thoughtful review.

---

### Official Review · Reviewer_k6XA · 2024-07-16

**Soundness:** 3
**Presentation:** 3
**Contribution:** 3
**Rating:** 6
**Confidence:** 2

**Summary:**

Recently, some studies have been exploring new optimization methods that can replace backpropagation, and one of the popular methods is direct feedback alignment (DFA). This paper adapts DFA to graph neural networks (GNNs). It replaces the gradient with a randomly initialized matrix multiplying the prediction error, so that we do not need to use backpropagation to update the parameters. We can directly update the model parameters based on the prediction error. Besides, for semi-supervised graphs where the label is scarce, we only have prediction errors for a few nodes, so the parameter update is not that effective. To overcome this drawback, the paper assumes that nearby nodes have similar prediction errors, so it uses a smoothing technique to generate pseudo prediction errors for unlabeled nodes.

**Strengths:**

(1) The studied problem is very interesting. The non-backpropagation optimization method is a very new field.

(2) The method design is good. Though it is an adaptation of direct feedback alignment (DFA) which has been used for Euclidean data, since this area has been rarely studied, I think the adaptation is also a contribution.

(3) There are some theories supporting the method.

(4) Extensive experiments verify its effectiveness, and it can outperform backpropagation.

**Weaknesses:**

(1) In Formula (4), it ignores the activation function when calculating the parameter update. Since the activation function is ReLU, ignoring it might not cause a big problem. However, when we apply other activation functions, or when we use a complex GNN architecture, I am afraid this method might not work as well.

(2) From Table 1, we can see that DFA achieves the best performance, while backpropagation is also very good on some datasets. I encourage the authors to add some discussions about when the proposed method is more useful than backpropagation, on what kinds of datasets, and what kinds of GNN models. Besides, the authors explained why DFA is better than previous non-backpropagation methods in Lines 254-274, but it would be better if the authors could explain why DFA is better than backpropagation.

(3) The authors can consider introducing a little more about the limitations of backpropagation and the non-backpropagation optimization methods in the introduction or related work. Since most readers are not familiar with non-backpropagation optimization methods, it deserves more introduction.

(4) The authors can improve some notations and make them clearer. For example, in Formula (5), the multiplication of $B^{(2)}$ and $B^{(1)}$ is written as $B^{(1)}$.

**Questions:**

(1) What is $W$ in Formula (7)?

(2) In Section 4.3, the authors point out that W will converge to be similar to B. Since B is randomly initialized, if B is at a high point in the loss landscape, then will the optimization fail? Is the randomness of B important to the performance?

(3) I might have not fully understood the intuition behind DFA. I am confused about why DFA performs better than backpropagation. I look forward to intuitive explanations.

(4) Table 6 shows that all the non-backpropagation methods require more time than backpropagation. If I have correctly understood, DFA can directly calculate the parameter update for every layer from the loss, so it does not need to calculate the parameter update layer by layer as in backpropagation, then why does it need more time than backpropagation?

**Limitations:**

The authors have discussed the limitations. I encourage the authors to further explain what kinds of datasets and what kinds of GNN models DFA is most suitable for.

---

> ### Author Rebuttal · Authors · 2024-08-04
>
> Thank you for your constructive feedback and positive assessment of our work. Below, we provide individual responses to your questions.
>
> $\textbf{\large Response to W1:} $
>
> We take the GCN with ReLU activation function as an example to elaborate on our method since GCN is one of the most classic GNN models and ReLU is the most popular activation function.
>
> Considering another activation function $\sigma(\cdot)$ with a deterministic derivative function $\sigma^{'}(\cdot)$, Eq.4 can be modified as: $\delta\textbf{X}^{(2)}=\textbf{S}^{\text{T}}\textbf{EB}^{(2)}\odot\sigma'(\textbf{H}^{(1)}\textbf{W}^{(1)})$ and $\delta\textbf{X}^{(1)}$ is approximated as $(\textbf{S}^{\text{T}})^{2}\textbf{EB}^{(1)}\odot\sigma'(\textbf{H}^{(0)}\textbf{W}^{(0)})$, which is similar to DFA in MLPs. The symbol $\odot$ denotes element-wise multiplication, and $\sigma'(\textbf{H}^{(k)}\textbf{W}^{(k)})$ is computed during forward propagation. As shown in **Tab.VIII** in the supplementary PDF, our method is well integrated with different activation functions (e.g., Sigmoid, Tanh, ELU and LeakyReLU) and derives consistently excellent performance.
>
> Regarding different GNN architectures, our method is portable to complex GNNs. Once $\delta\textbf{X}^{(k)}$ is derived in parallel, the parameters of each GNN layer can be computed by gradient descent. This adaptability ensures that our method remains effective while still eliminating the backpropagation of gradients among different GNN layers. The results in Tab.3 of Sec.5.4 demonstrate the excellent portability of our method.
>
> $\textbf{\large Response to W2 and Limitations:} $
>
> Thanks for your suggestion, and we will add more discussion in the revised version.
>
> It is observed from Tab.1 that our method has overall superior performance than BP. In particular, our method shows a much more significant accuracy improvement on relatively small datasets, such as Texas and Cornell. The reason mainly lies in that GNNs trained by BP often suffer from heavy overfitting due to the scarcity of supervision on small datasets, especially for semi-supervised tasks. By contrast, the pseudo error generated in our method not only is for gradient computation when adapting DFA to GNNs, but also serves as a data augmentation strategy, enhancing the robustness of GNNs (shown in Fig.3) and improving the ability for handling overfitting with scarce supervision. The gains derived from the pseudo-error strategy is not so significant on larger datasets (e.g., PubMed, Photo), but it still brings overall positive effects to our methods. Interestingly, BP does not benefit from the same pseudo-error generation strategy (please see **Tab.IX** in the supplementary PDF, and refer to $\textbf{the response to reviewer buRu}$ for detailed discussion).
>
> Additionally, from Tab.3, it is observed that our method shows greater superiority when the aggregation strategy of the GNN models is isotropic or the receptive field of aggregation is larger (e.g., GCN, SGC and ChebyNet). We attribute this phenomenon to the fact that isotropic and larger receptive field GNN layers can better utilize the direct feedback of errors. This is an interesting finding that deserves further exploration.
>
> $\textbf{\large Response to W3:}  $
>
> Thanks for the suggestion. We will introduce more about the limitations of BP and the non-BP optimization methods in the introduction and related work in the revised version.
>
> $\textbf{\large Response to W4:} $
>
> Thanks for your suggestion, we will provide notations for this variable in revision.
>
> $\textbf{\large Response to Q1:}  $
>
> Sorry for the typo, the $\textbf{W}$ in Eq.7 should be $\textbf{Z}$, which is the variable to be optimized.
>
> $\textbf{\large Response to Q2}  $
>
> Sorry for the confusion caused. The alignment of $\textbf{W}$ and $\textbf{B}$ means the directions of the two variables tend to converge to within 90 degrees of each other (as shown in Fig.2 (a)) rather than being numerically similar. In fact, the initialization of  $\textbf{B}$ is not completely random, but follows a normal distribution that satisfies $\mathbb{E}[\textbf{B}\textbf{B}^{T}]\propto \textbf{I}$. Under such constraints, as discussed in Sec. 4.3, $\textbf{W}$ will converge to be alignment to $\textbf{B}$ even starting from a high point.  We will add related description in the revised version for clarity.
>
> $\textbf{\large Response to Q3:}  $
>
> DFA is BP-free training method that has not been fully explored yet. Intuitively, DFA updates model parameters via gradient computation that is similar to BP. But differently, DFA obtains approximate gradients directly from the loss rather than backpropagation layer-by-layer. As discussed in Sections 1 and 2, this difference makes DFA more biologically plausible and promising in terms of generalization and parallelism. In previous studies, DFA usually exhibits similar or inferior performance in comparison with BP. The performance improvement of our method mainly derives from the pseudo-error generation strategy, which has been discussed in the response to W2.
>
> $\textbf{\large Response to Q4:}  $
>
> Yes, DFA can directly calculate the parameter update for every layer from the loss, and it thus can improve training efficiency by parallelization.  However, to generalize DFA to graph data we introduce the pseudo-error generator as elaborated in Sec.4.2, and the main time cost is from the pseudo-error generation (Eqs.7, 8) and node filtering (Eq.9). According to Eqs.7 and 8, the time complexity of the two parts is $\mathcal{O}(tcn^{2})$ and $\mathcal{O}(cn)$ respectively, where $t, c, n$ denote the iteration epochs for error spreading, the number of categories and the number of nodes, respectively.

---

> > ### Comment · Reviewer_k6XA · 2024-08-12
> > **Reply to the rebuttal**
> >
> > Thank you for your reply! Your responses have addressed most of my concerns. The authors conducted a lot of experiments to verify the proposed method from various perspectives. Nonetheless, it seems that this method has some limitations regarding its suitable scenario and training efficiency that are worth further discussion in the paper. It would be better if the authors could include more discussions about the insights and limitations in the paper. I would like to maintain my score.

---

> ### Author Response · Authors · 2024-08-12
>
> Thank you for your insightful feedback and for recognizing our efforts. We appreciate your suggestion to further discuss the method's limitations, particularly regarding its suitable scenarios and training efficiency.
>
> During the rebuttal period, we conducted additional experiments to address these concerns, focusing on the method's suitability for various activation functions, its performance on large-scale graphs, training time, and convergence behavior. These insights have provided a more comprehensive understanding of the method's capabilities and constraints. In the final version of the paper, we will include a dedicated section discussing these aspects, offering a balanced view of the method's strengths and areas for future improvement.
>
> We respect your decision and are grateful for your constructive critique. Your feedback has been instrumental in refining our work, and we thank you once again for your thorough and thoughtful review.

---

### Author Rebuttal · Authors · 2024-08-05

$\definecolor{mybgcolor}{RGB}{249,242,244}$
$\definecolor{Flickr}{RGB}{140,27,19}$
$\definecolor{Reddit}{RGB}{140,27,19}$
$\definecolor{ogbn-arxiv}{RGB}{140,27,19}$
$\newcommand{\highlight}[2]{\colorbox{#1}{\textcolor{#2}{#2}}}$

Dear ACs, PCs and all reviewers,

We would like to express our gratitude to all the reviewers for their valuable comments and feedback on our work. All reviewers recognized the soundness, presentation and quality of our manuscript. We have responded to all the questions posed by each reviewer. In this summary, we aim to provide you with a clear understanding of the changes made during the rebuttal process.

$\textbf{\large 1. Inclusion of large-scale graph datasets for experiments}$

In response to the insightful suggestions from reviewer **mUbj**, we discuss the portability of our method to large graph datasets and conduct additional experiments on three new large-scale graph datasets the reviewer suggested. $\textbf{Table X}$ in the supplementary PDF displays the results on $\highlight{mybgcolor}{Flickr}$, $\highlight{mybgcolor}{Reddit}$ and $\highlight{mybgcolor}{ogbn-arxiv}$ in comparison with seven baseline algorithms. The results demonstrate our method outperforms all the non-BP methods and has comparable performance to BP (within 1\% accuracy), with only a little memory cost.

$\textbf{\large 2. More extensive ablation studies}$

- As requested by reviewer **buRu**,  we plot the training and validation accuracy of the proposed method and BP over training epochs on three datasets to demonstrate the convergence of our method as shown in $\textbf{Fig.4}$ in the supplementary PDF. The results show our method has similar convergence compared with BP. Additionally, we provide a detailed discussion of the figures to highlight the training effectiveness of our method.

- As requested by reviewer **k6XA**, to demonstrate that our method is feasible with different activation functions other than ReLU, we conduct experiments for our method with four different activation functions (i.e., Sigmoid, Tanh, ELU and LeakyReLU) as shown in $\textbf{Tab.VIII}$ in the supplementary PDF. The results demonstrate our method is well integrated with different activation functions and derives consistently excellent performance.

- As requested by reviewer **buRu**, we plot the prediction accuracy with the number of layers increasing from 2 to 10, as shown in $\textbf{Fig. 5}$ in the supplementary PDF. Both the BP and our method show a decrease in accuracy with the number of layers increasing due to oversmoothing. However, our method is less negatively affected and consistently outperforms BP. This superiority is derived from the fact that our method adopts direct feedback of errors for the update of each layer, reducing the problem of oversmoothing and other gradient issues caused by backward gradient accumulation. This observation indicates the potential of our method for deep GNN training.

- As requested by reviewer **buRu**, we integrate the proposed pseudo-error spreading component into BP. The experimental results from $\textbf{Tab.IX}$ in the supplementary PDF show that although this component contributes to DFA in our method, it does not positively enhance BP overall. It indicates that DFA is more robust against noise in generated pseudo error than BP, which is consistent with the analysis in some previous studies.

$\textbf{\large 3. Algorithm complexity analysis}$

In response to the insightful suggestions from reviewer **mUbj**, we provide a detailed analysis of the time complexity of our method. We present the time complexity of each component and demonstrate that our method has the same order of time complexity as BP. To apply our method to large datasets, we explain how to reduce the space requirement of graph information aggregation from $\mathcal{O}(n^{2}+nd)$ to $\mathcal{O}(|\mathcal{E}|+nd)$ through algorithm optimization.

$\textbf{\large 4. Highlighting the superiority compared with BP and non-BP methods}$

- As requested by reviewer **k6XA**, we provide a discussion about when our method is more useful than BP. We point out that the superiority of our method is more significant than BP when the dataset scale is relatively small,  the GNN model is isotropic, or the receptive field of aggregation is large.

- As requested by reviewer **k6XA**, we provide an intuitive explanation of why our method outperforms BP. We attribute the superior performance of our method to the data augmentation from our well-designed pseudo-error generator and the robustness of the DFA mechanism to noisy augmentation.

- As requested by reviewer **mUbj**, we provide an explanation of why our method shows a notable variation in the performance improvements observed across different datasets.

We want to express our gratitude to the reviewers for their valuable suggestions and would greatly appreciate any further questions they may have regarding our paper. If our responses have well-addressed your questions, we would kindly ask for your reconsideration of the scores.

---

### Decision · Program_Chairs · 2024-09-25

**Decision:**

Accept (poster)

**Comment:**

This paper receives overall positive reviews and the authors have actively engaged in rebuttals. One of the reviewers was concerned/confused with the computational complexity, and the authors have been trying to actively address them during rebuttals. However, this reviewer did not provide a final confirmation as to if the concerns have been addressed. I did a check and believe the concern has been properly addressed. Thus an accept is recommended.